# Changes in Gut Microbiota, Midgut Structure, and Gene Expression of *Spodoptera frugiperda* Infected by *Serratia marcescens*

**DOI:** 10.3390/insects16090933

**Published:** 2025-09-04

**Authors:** Yibo Guo, Yue Zou, Youyang Chen, Jiaxin Liu, Yingying Ye, Xinglong Huang, Zhengwei Wu

**Affiliations:** 1Department of Agronomy, College of Coastal Agricultural Sciences, Guangdong Ocean University, Zhanjiang 524088, China; gyb3824@163.com (Y.G.); gq121512@126.com (Y.Z.); 13169112583@163.com (Y.C.); 18690891172@163.com (J.L.); 13724323733@163.com (Y.Y.); 2South China Center, National Center of Technology Innovation for Saline-Alkali Tolerant Rice, Zhanjiang 524088, China; 3Hunan Provincial Key Laboratory of Ecological Conservation and Sustainable Utilization of Wulingshan Resources, College of Biology and Environmental Sciences, Jishou University, Jishou 416000, China; hxl4645@163.com

**Keywords:** *Serratia marcescens*, *Spodoptera frugiperda*, insecticidal activity, midgut structure, microbial community, transcriptome analysis

## Abstract

This study investigated the lethal mechanism of *Serratia marcescens* in *Spodoptera frugiperda,* the fall armyworm (FAW), at both the molecular and cellular levels. By analyzing the midgut structure, hemocyte composition, gut microbiota, and transcriptomic data of the FAW, we found that *S. marcescens* infection disrupts the midgut integrity of the fourth instar larva of the FAW, leading to systemic sepsis and ultimately death. Our findings elucidate the pathogenic mechanism of *S. marcescens* against the FAW and provide insights for the development of novel biological control strategies.

## 1. Introduction

The fall armyworm (FAW), *Spodoptera frugiperda*, is native to the Americas and invaded China in 2018 [1]. The FAW is a highly adaptable, migratory, and destructive pest that poses significant threats to crops due to its polyphagous and voracious characteristics. In response to the rapid spread of the FAW and its threat to food and livelihood security in many parts of the world, the FAO launched a pioneering global FAW prevention and control initiative in December 2019. In China, it can cause year-round damage in the southern provinces [2]. In terms of control, biological control has emerged as a primary focus of research alongside common physical and chemical methods. In places like Brazil and Nepal, the release of parasitic wasps has been proven to reduce the number of fall armyworms in the fields [3]. Additionally, a variety of pathogenic microorganisms, such as bacteria, fungi, and viruses, have been detected in FAW [4]. For instance, Bacillus thuringiensis (Bt), a Gram-positive bacterium residing in soil, is widely employed as a biopesticide to manage insect pests, including FAW, by producing crystal proteins called delta-endotoxins with inherent insecticidal properties [5,6]. However, there is only a limited number of Bt products proven effective against FAW that are available in the market for controlling lepidopteran pests [7].

*Serratia marcescens*, a member of the phylum Proteobacteria and family Enterobacteriaceae, is a bacterium commonly found in soil, water, and the intestines of insects [8]. This bacterium is known for its production of extracellular enzymes, such as chitinases, which are particularly relevant due to the fact that chitin serves as a primary component of the insect cuticle and the peritrophic membrane (PM), a protective covering in the digestive tracts of numerous insects [9,10]. *S. marcescens* chitinases could degrade the PM and gut lining, linking to pathogenicity. The chitinases and proteases produced by *S. marcescens* have the potential to function as larvicides targeting Anopheles mosquitoes [11]. The midgut epithelium of insects contains a chitinous layer that contributes to their ability to adapt to environmental conditions [12]. The extracellular products of *S. marcescens* may have significant effects on the insect midgut, while insect microbial symbionts are capable of metabolizing host plant defense chemicals, attenuating or suppressing plant responses, and providing hosts with essential nutrients or digestive processes [13,14]. However, in current studies, the effect of chitinase produced by *S. marcescens* on the midgut of FAW is still rarely reported. The strain *S. marcescens* has been reported to have insecticidal activity against FAW [15].

The midgut plays a crucial role in the insect digestive system as a key site for food conversion into energy [16]. Additionally, it is involved in cellulose metabolism and facilitating the acquisition of vitamins, proteins, and various nutrients by the host insect [17,18]. The lepidopteran midgut is the second largest organ. In Lepidoptera, in addition to secreting digestive enzymes, the microbial community within the midgut plays a crucial role in maintaining a stable intestinal environment, improving nutrient acquisition efficiency, and providing resistance and immunity against foreign pathogens [19,20,21].

In this study, we determined the virulence of the fermented bacterial solution of *S. marcescens* against the egg, larval, and pupal stages of FAW. Additionally, we investigated the changes in the midgut tissue and midgut microbial community of FAW following infestation by *S. marcescens*. Furthermore, we investigated the molecular effects of *S. marcescens* on the midgut tissue of FAW through transcriptome sequencing analysis. The objective of our research is to offer valuable insights into the lethal mechanism of *S. marcescens* on FAW and to establish a theoretical foundation for pest control and agricultural applications.

## 2. Materials and Methods

### 2.1. Experimental Materials

FAW were reared in a climatic chamber and constituted the 32nd generation of an indoor population. The larvae were fed corn leaves from the Zhengdan 958 variety cultivated in the greenhouse of Guangdong Ocean University with environmental conditions maintained at a temperature of 28 °C and a photoperiod of 16 L: 8 D.

*S. marcescens* ZJ9 (isolated from mangrove tidal flat soil) has been preserved in our laboratory, with the original strain maintained at Guangdong Ocean University. Additionally, a backup strain is deposited at the Guangdong Microbial Culture Collection Center under catalog number 1.4235. Prior to use, a small aliquot of the mixed bacterial suspension was withdrawn from the tube and subjected to streak plating on LB agar to isolate contamination-free single colonies. These colonies were subsequently cultivated in LB liquid medium. The activation process was carried out at 28 °C with shaking at 180 rpm for 24 h, leading to the preparation of the mother liquor (ML). This ML was subsequently inoculated onto slant LB medium and LB solid medium for future use. The slants and plates were incubated at 28 °C until sufficient growth was observed, after which they were stored at 4 °C in a refrigerator. Bacteria were routinely recovered from the LB slant or plate media prepared from the ML.

### 2.2. Toxic Effects of S. marcescens on FAW

After activating the prepared strain, the bacterial solution was diluted with phosphate-buffered saline (PBS) to a concentration of 0.01 mol/L, using the 10× PBS concentrate prepared on-site. The LB medium was formulated with 10 g/L of tryptone, 5 g/L of yeast extract, and 10 g/L of NaCl, supplemented with 15 g/L of agar for solid culture medium. Bacterial cultures were obtained from the plate or slant medium as previously described, and single colonies were isolated through plate streaking. These isolated colonies were then incubated in LB liquid medium at 28 °C and 180 rpm for 36 h. For the virulence test, the bacterial solution was diluted with PBS and supplemented with sterilized Tween 80 to achieve a final concentration of 0.10%. Select plates with a colony count ranging from 30 to 300 for counting (plates with a colony count less than 30 or greater than 300 have a relatively large error and need to be excluded). If multiple dilutions fall within the range, take the average value. Bacterial counts were conducted using the diluted plate smear method, and five concentrations of 1 × 10^5^, 1 × 10^6^, 1 × 10^7^, 1 × 10^8^, and 1 × 10^9^ CFU/mL were obtained, which were used for virulence determination tests. The calculation formula is:Original sample CFU/mL = Number of colonies on the plate × (1/coating volume) × dilution factor

Oocyte virulence determination: Several oocyte masses of the same size were taken from the same batch and immersed in the prepared bacterial solution of different concentrations for 30 s. Three repetitions for each concentration gradient were set up, as were 30 oocyte masses for each repetition. Subsequently, the egg blocks were transferred to the artificial climate chamber. The number of hatched eggs was counted daily, and the hatched larvae were transferred in a timely manner until no more larvae hatched. The unhatched eggs were counted, and the hatching mortality rate was calculated.

Optimal soil moisture for emergence: The same batch of fall armyworms with newly hardened pupal shells was selected for the experiment. Seven concentration gradients with a relative moisture content ranging from 10.00% to 70.00% were set, and the mixture was prepared using loess pre-sieved out of 200-mesh soil and double-distilled water. The prepared yellow soil was carefully sprinkled into the uncovered 15 mL dipping cup for the pupae, then the dipping cup was placed into a 100 mL small beaker and sealed with a rubber band and a breathable sterilizing sealing film. Three repetitions were set for each humidity level, with 30 heads in each repetition. Each beaker was weighed, and the weight lost due to water evaporation was supplemented with a pipette every 24 h. The pupae were observed using tweezers to hold them and observe their survival, and the daily emergence of adult pupae was counted for a total of 30 days. Deformed adult pupae and those that failed to emerge successfully within 30 days were judged as dead.

Pupal stage virulence determination: Tweezers were used to place the same batch of fall armyworms with freshly hardened pupal shells into the bacterial solution, and they were shaken evenly for 30 s before taking them out. The control group was replaced with 0.50% Tween 80 solution. Thirty heads were set each time. The pupae of the fall armyworms were soaked in diluted bacterial solution for 30 s, transferred to sterilized uncovered dipping cups, and loess was prepared using soil moisture with the highest relative humidity survival rate. The treatment was as described above. The daily emergence of adult insects was statistically recorded for a total of 30 days. Deformed adult insects and those that failed to emerge successfully within 30 days were judged as dead.

Larval virulence determination: The fourth-instar larvae of the same generation of FAW were selected and set aside after 12 h of starvation. Each group consisted of 30 insects, with 3 replicates. Fresh corn leaves were soaked in the bacterial solution to be tested for 15 s and then dried. Each insect was fed 8 leaves carrying the bacteria (12 × 12 mm), and then the corn leaves were supplemented without the bacterial solution every 24 h. The survival status was checked every 12 h. If there was no response when touched lightly with a glass rod, the insect was considered dead and recorded. The mortality rates and adjusted mortality rates of the test insects at 12, 24, 36, 48, 60, and 72 h under different concentrations of bacterial solutions were respectively calculated. The corrected mortality rates were converted into probability values, and probability analyses were conducted after converting the concentrations into pairs. The Karber method was used to obtain the virulence regression equation, and the median lethal concentration (LC50) was calculated. SPSS 26 was used to analyze the data. The calculation formula is as follows:Mortality rate/% = Number of dead test insects/total number of tested insects × 100;Corrected mortality rate/% = (treatment group mortality rate − control group mortality rate)/(1 − control group mortality rate) × 100.

### 2.3. Effect of S. marcescens on the Hemolymph of FAW Larvae

Fourth-instar larvae with consistent physiological states were selected. After 4 h of starvation treatment, samples were collected at 24 h, 48 h, and 72 h. Each time point was repeated three times, and 10 larvae were collected each time. The larvae were anesthetized on ice for 15 min (with reduced activity), then rinsed with sterile water and dried. Small incisions were made on the third pair of thoracolegs, and hemolymph was collected using a 10 μL pipette and stored in a centrifuge tube. After staining with Wright-Giemsa, blood cells were classified and counted under a microscope. For each larva, three slides were prepared. Multiple fields were selected for each slide, with approximately 300 cells in each field. The average value was taken for statistics. The SPSS 26 software was used to conduct single-factor ANOVA analysis for the control group and the treatment group with the same blood cells at the same time and type.

### 2.4. Determination of Changes in the Midgut Microbial Community in FAW

Larvae hatched from egg masses of the same pair of adults were reared until the second instar before being isolated in 15 mL cups and fed on tender corn leaves. The leaves were cut into circular disks using a 12 mm punch for the feeding trials. The treatment group utilized fresh corn leaves treated with a diluted bacterial solution of optical density 1.2, while the control group used leaves treated with a 0.10% Tween 80 solution. Two hundred fourth-instar larvae of uniform growth were selected for the experiment. Each feeding cup contained eight leaf disks after 12 h of starvation, which were replenished daily with fresh leaves for three days. The larvae were starved for 12 h before feeding. After feeding, only the processed leaves were given to them. Each cup contained one larva, which was isolated in the cup. Sampling times were set at 24 h, 48 h, and 72 h with three replicates per time point, and ten larvae were randomly sampled per replicate. The larvae were sterilized with 75.00% ethanol and then washed with sterile water. Dissections were performed on an ultra-clean workbench, with whole larvae placed on ice and quickly dissected along the ventral midline to expose the digestive tract. The digestive tract was then excised, washed with PBS, and stored in sterile centrifuge tubes before being flash-frozen in liquid nitrogen. Total genomic DNA was extracted from the above samples using the TGuide S96 Magnetic Soil/Stool DNA Kit (Tiangen Biotech (Beijing) Co., Ltd., Beijing, China) according to the manufacturer’s instructions. The hypervariable region V3-V4 of the bacterial 16S rRNA gene was amplified with primer pairs 338F: 5′-ACTCCTACGGGAGGCAGCA-3′ and 806R: 5′-GGACTACHVGGGTWTCTAAT-3’. PCR products were checked on agarose gel and purified through the Omega DNA purification kit (Omega Inc., Norcross, GA, USA). The purified PCR products were collected, and paired-end sequencing (2 × 250 bp) was performed on the Illumina Novaseq 6000 platform (Illumina, San Diego, CA, USA).

Alpha diversity analysis was performed to identify the complexity of species diversity of each sample utilizing QIIME2 software. Beta diversity calculations were analyzed by principal coordinate analysis (PCoA) to assess the diversity in samples for species complexity. One-way analysis of variance was used to compare bacterial abundance and diversity. Linear discriminant analysis (LDA) coupled with effect size (LEfSe) was applied to evaluate the differentially abundant taxa. The online platform BMKCloud (https://www.biocloud.net) was used to analyze the sequencing data.

### 2.5. Midgut Tissue Sectioning and Observation

The procedures for midgut processing and sampling followed the same protocol as in the determination of midgut microbial community changes in FAW. Alpha diversity analysis was performed to identify the complexity of species diversity of each sample utilizing QIIME2 software; 16s sequencing was used. The online platform BMKCloud (https://www.biocloud.net) was used to analyze the sequencing data. The excised midgut was fixed in 2.50% glutaraldehyde for 24 h and sent to Hangzhou Yanqu Information Technology Co., Ltd., Hangzhou, China for sectioning and imaging. Sections were prepared using the paraffin embedding method, with a section thickness of 4 µm, and stained with hematoxylin and eosin (H&E). The sections were observed using an optical microscope with a magnification of 400× (usually a 40× objective lens combined with a 10× eyepiece), which can clearly present the fine structure of the cells.

### 2.6. Transcriptomic Sequencing of FAW Infected with S. marcescens ZJ9

Total RNA was extracted according to the instruction manual of the TRlzol Reagent (Life Technologies, Carlsbad, CA, USA), with a total of 18 samples. RNA concentration and purity was measured using NanoDrop 2000 (Thermo Fisher Scientific, Wilmington, DE, USA). RNA integrity was assessed using the RNA Nano 6000 Assay Kit of the Agilent Bioanalyzer 2100 system (Agilent Technologies, Santa Clara, CA, USA). A total amount of 1 μg RNA per sample were used as input material for the RNA sample preparations. Sequencing libraries were generated using the Hieff NGS Ultima Dual-mode mRNA Library Prep Kit for Illumina (Yeasen Biotechnology (Shanghai) Co., Ltd., Shanghai, China), and index codes were added to attribute sequences to each sample. mRNA was purified from total RNA using poly-T oligo-attached magnetic beads. First-strand cDNA was synthesized, and second-strand cDNA synthesis was subsequently performed. Remaining overhangs were converted into blunt ends via exonuclease/polymerase activities. After adenylation of 3’ ends of DNA fragments, NEBNext adaptors with hairpin loop structures were ligated to prepare for hybridization. The library fragments were purified with the AMPure XP system (Beckman Coulter, Beverly, MA, USA). Then 3 μL USER Enzyme (NEB, Ipswich, MA, USA) was used with size-selected, adaptor-ligated cDNA at 37 °C for 15 min followed by 5 min at 95 °C before PCR. Then PCR was performed with Phusion High-Fidelity DNA polymerase, universal PCR primers and the index (X) primer. At last, PCR products were purified (AMPure XP system), and library quality was assessed on the Agilent Bioanalyzer 2100 system. The libraries were sequenced on an Illumina NovaSeq platform to generate 150 bp paired-end reads. The raw reads were further processed with the BMKCloud (www.biocloud.net) online platform.

The handling and sampling method for the FAW was consistent with the determination of midgut microbial community changes in FAW, with sampling times set at 24 h and 48 h post-treatment. Transcriptomic sequencing was commissioned to Biomarker Technologies, and differential gene expression analysis was carried out using DESeq2 for each replicate; three biological replicates were used for each condition. Differentially expressed genes were identified based on a log_2_ fold-change ≥ 1 and FDR < 0.01 as the selection criteria. DEGs were identified with DESeq2 using |log2FC| ≥ 1 and FDR < 0.01. Data analysis was performed using the Biomarker eukaryotic reference genome transcriptome analysis platform.

Quality control: Raw data (raw reads) of fastq format were firstly processed through in-house Perl scripts. In this step, clean data (clean reads) were obtained by removing reads containing adapters, reads containing poly-N and low-quality reads from raw data. At the same time, Q20, Q30, and GC content and the sequence duplication level of the clean data were calculated. All the downstream analyses were based on clean data with high quality.

Gene annotation was carried out using databases such as the Gene Ontology (GO) Database (http://www.geneontology.org/) and the Kyoto Encyclopedia of Genes and Genomes (KEGG) Database (http://www.genome.jp/kegg/). Software InterProScan 5.34-73.0 was used for GO annotations. The statistical enrichment degree of differentially expressed genes in the KEGG pathway was detected using the clusterProfiler 4.4.4 software.

### 2.7. The qRT-PCR

The qRT-PCR was used to verify the expression levels of randomly selected genes with the internal reference gene RPL13 (*LOC118279579*) [22]. The results obtained from the transcriptome analysis and identification of significantly differentially expressed genes in both control and treated groups were utilized for validation. Retrotranscription was performed using the EvoM-MLV premix kit, with a reaction system consisting of 20 µL total volume, comprising 10 µL RNA, 4 µL 5X Evo M-MLVRT Mix Ver.2, and 6 µL RNase-free water. The reaction conditions included incubation at 37 °C for 15 min followed by denaturation at 85 °C for 5 s.

After reverse transcription of the original RNA, the qRT-PCR system was performed using 2X SYBR Green Pro Taq HS Premix IV (10 µL), cDNA (2 µL), Primer F/R (10 µM)2 (2 µL), and RNase-free water (6 µL). The qRT-PCR kit used was the SYBR Green Pro Taq HS premixed qPCR kit. Primers are detailed in Table 1, and 10 genes with a log2FC value greater than 2.0 were randomly selected for validation based on the transcriptome results. Reference genes were selected for RPL 13, and the procedure involved a reaction at 95 °C for 30 s once, followed by cycling at 95 °C for 30 s, then at 55 °C for another 30 s, and finally at 72 °C for another cycle of 30 s repeated forty times. The final step is to be saved at 4 °C. The amplification and dissolution curves of the raw data were analyzed using the original built-in software, CFX Manager^TM^ 3.1 Softwore (Bio-Rad Laboratories, Inc., Hercules, CA, USA), with three templates for each target gene, each in triplicate. The calculation of the relative expression level of genes adopted the ΔΔCt method. The specific formula is as follows: Relative expression level = 2^−ΔΔCt^, where ΔCt = Ct value of the target gene − Ct value of the internal reference gene, and ΔΔCt = ΔCt value of the experimental group − ΔCt value of the control group. *t*-tests were performed using SPSS 26 software, and GraphPad Prism 8 was used for confirmation.

## 3. Results

### 3.1. Toxic Effects of S. marcescens on FAW

The results of treatment at the egg stage (Figure 1a) indicated a direct correlation between the concentration of *S. marcescens* and its lethal effects on FAW eggs, indicating a positive relationship between bacterial solution concentration and lethality. *S. marcescens* activity depends on moisture. The study revealed that at a soil relative water content of 30.00%, there was an optimal eclosion rate for pupae (Figure 1b). It was observed that the presence of *S. marcescens* in the soil exhibited a positive correlation with the mortality rate among pupae (Figure 1c). We observed a significant rise in pupal mortality above 1 × 10^7^ CFU/mL, with an estimated LC50 of 7.50 × 10^7^ CFU/mL. Furthermore, by establishing a linear regression equation, the half-lethal concentration (LC50) of *S. marcescens* suspension to FAW larvae at different time points was determined; the larvae were the fourth instar larvae (Figure 1d). It shows the LC50 of *S. marcescens* ZJ9 for larvae at different time periods. LC50 is negatively correlated with the time point; this proves the dependence of LC50 on time. Following comparative analysis, it was concluded that a treatment concentration of 1 × 10^9^ CFU/mL would be utilized for subsequent experiments.

### 3.2. Effect of S. marcescens on the Hemolymph of FAW Larvae

After treatment, the FAW larvae exhibited a distinctive reddish-brown coloration, accompanied by a wrinkled and fragile cuticle. The hemolymph within the hemocoel was discernible through the epidermis as a dark reddish-brown hue (Figure 2a). In accordance with Koch’s postulates, *S. marcescens* was successfully isolated from the blood of the treated larvae; conversely, no such colonies were detected in the control group.

Blood samples were collected from both the control and treatment groups at 24, 48, and 72 h post-treatment to evaluate blood cell counts and morphology. In the control group, plasmatocytes (PLs) typically exhibited a round or oval shape, with some displaying a partial spindle shape (Figure 2b), often clustering together (Figure 2c). These cells were basophilic, showing a light blue coloration following staining. Granulocytes (GRs) presented various shapes, predominantly round, with some extending pseudopodia. After staining, these cells contained heterogeneous purplish-red lysosomal granules (Figure 2b). Spherulocytes (SPs) featured prominent bead-shaped inclusions at the periphery of the cell and displayed a central dark purplish-red chaotic mass post-staining (Figure 2a). In the treatment group, plasmatocytes and spherulocytes appeared similar to the controls. Many granulocytes in infected larvae extended pseudopodia (phagocytic projections), whereas control-group granulocytes remained mostly round. However, GRs in the control group were primarily circular, while those in the treatment group showed a notable proportion extending pseudopodia after infection (Figure 2a).

A notable increase in PLs was observed at 24 h post-infection, reaching 201.51% of the control group’s levels (*p* < 0.05) (Figure 3). This was followed by a decrease at 48 h (84.87% of the control group) and a subsequent rise at 72 h (108.21% of the control group). The GRs exhibited an increase of 41.89% compared to the control at 48 h, although it remained relatively lower during all other time points. The SPs significantly decreased at 24 h (*p* < 0.05), only achieving 21.11% of the control level, before continuing to rise and ultimately reaching levels comparable to those of the control thereafter. Overall, after *S. marcescens* infection, PLs in FAW hemolymph rise rapidly, SPs drop swiftly, and GRs fluctuate later.

### 3.3. The Influence of S. marcescens Infestation on the Midgut of FAW

The midgut sections of FAW, which exhibited partial mortality after feeding on *S. marcescens* ZJ9, were analyzed over three distinct time periods. Controls displayed intact peritrophic membranes (PM) and progressively thickened intestinal walls, with no observable signs of cellular damage (Figure 4a–c). In the larvae treated with ZJ9, significant changes were observed at 24 h: the midgut epithelium began to separate (gaps), and goblet cells elongated (Figure 4d). By 48 h the peritrophic membrane was mostly gone, with enlarged intercellular spaces forming vesicles. The number of goblet cells decreased, and their morphology became distorted (Figure 4e). After 72 h, the epithelium showed many regenerative cells, and the lumen was highly constricted. The peritrophic membrane, largely disrupted by 72h, revealed more pronounced alterations in midgut structure. These changes included an increase in regenerative cells, wrinkling of the midgut, and distortion of cellular morphology (Figure 4f). In addition, the treatment group exhibited a significant reduction in intestinal diameter (*p* < 0.05) (Figure 4j–l) and an increase in the number of intestinal wall cells, whereas no notable changes were observed in the control group (Figure 4g–i). This was measured according to the scale of a ruler, the 200-micron scale shown in the lower left corner of the figure, observed using an optical microscope. By day 3 post-infection, numerous intestines exhibited marked luminal narrowing (Figure 4k) and a wrinkled appearance (Figure 4l).

### 3.4. Changes in the Midgut Microbiota of FAW after Infection with S. marcescens

Cluster analysis revealed that the samples contained microorganisms from 2 kingdoms, 44 phyla, 105 classes, 267 orders, 545 families, 1210 genera, and 1535 species. Species clustering analysis identified 1394 bacterial species (predicted species) in the control group and 1229 in the treatment group, with 827 species shared between them (Figure 5). Alpha diversity analysis (Figure 6), based on OTU clustering (Figure 7), revealed a significant increase in microbial diversity in the treatment group. Alpha diversity (Shannon index) initially dropped in treated larvae (24–48 h), but by 72 h the gut community was significantly more diverse than controls.

Genus-level analysis revealed a significant increase in the relative abundance of Serratia at 24 h post-treatment. Although this abundance gradually declined thereafter, it remained consistently higher than that in the control group across all time points (Figure 7). The highest *Serratia* abundance in the midgut occurred at 24 h, decreasing substantially by 72 h (Figure 8). Concurrently, the proportion of “Others” (defined as the total sum of bacteria with a relatively low proportion and a relative abundance of less than 1.00%) increased obviously. The introduction of *S. marcescens* significantly altered the temporal dynamics of midgut microbial diversity (Figure 7), resulting in higher diversity in the treatment group compared to the control at 72 h.

LefSe analysis identified marker microorganisms in both the control and treatment groups. *Serratia* species exhibited significantly higher discriminatory power in the treatment group, while Enterococcus showed the highest contribution in the control group. Enterococcus was significantly enriched in control guts, whereas *Serratia* dominated the treated group (Figure 9). The LDA scores further revealed significant enrichment of specific bacterial taxa in the treatment group following *S. marcescens* ZJ9 introduction. Furthermore, random forest analysis (500 cycles) demonstrated that *Serratia* exerted the strongest influence on the midgut microbial community in the treatment group, contrasting with Enterococcus’s dominant role in controls. Notably, *Serratia* ranked fourth in overall feature importance (Figure 10). The top three are, in order: “f_Yersiniaceae”: Its LDA score is the highest in the green (Treat group), and it performs outstandingly in importance dimensions such as inter-group differences, making it a top-ranked group. “s__unclassified_*Serratia*”: Also has a high LDA score in the Treat group, ranking high in importance. “o__Flavobacteriales” also had a high LDA score in the Treat group, ranking third in terms of comprehensive importance. Through significant inter-group differences and other factors, they became the top three important groups, jointly demonstrating the differential characteristics of the microbial communities in the Treat group and the CK group (red) with *Serratia*.

### 3.5. Assembly of Fall Armyworm Transcriptome Data

Analysis of differential gene expression revealed a significant increase in the number of differentially expressed genes (DEGs; |log2FC| ≥ 1) at both post-treatment time points compared to their respective control groups (Figure 11). When FC ≥ 2, the expression level or abundance of the target gene or microbial group in the experimental group is at least twice that of the control group, showing upregulation. Notably, At 24 h after infection, we found that genes E, K, C, and M were upregulated, while genes A, B and D were downregulated; at 48 h, these increased to 256 and 244. The total number of DEGs escalated substantially over time, reaching a level at 48 h that was 2.6 times higher than the count observed at 24 h. This temporal increase was driven by both up- and down-regulated genes, with down-regulated DEGs increasing 2.02-fold and up-regulated DEGs showing a more pronounced increase of 3.61-fold between the 24-h and 48-h time points.

### 3.6. Functional Annotation Analysis of Differentially Expressed Gene COG

Upregulated genes were enriched in energy production (COG category C), amino acid metabolism (E), transcription (K), and cell wall/membrane biogenesis (M) (Figure 12a,b). Downregulated genes were enriched in RNA processing (A), chromatin structure (B), and cell cycle (D). Conversely, downregulated genes were enriched in RNA processing (A), chromatin structure (B), and cell cycle (D).

At 48 h (Figure 12c,d), COG annotation showed continued upregulation of energy production and conversion (C) genes and persistent enrichment of cell wall/membrane biogenesis (M). However, decreased gene frequency was specifically observed in coenzyme transport and metabolism (H) and nucleotide transport/metabolism (F) categories. Notably, transcriptome data revealed sustained downregulation of nucleotide transport/metabolism (F). Combined with histological observations demonstrating severe midgut structural disruption post-infection, these findings align with the significant upregulation of signal transduction mechanisms (T) in the transcriptome. The significant upregulation of signal transduction mechanisms (T) is of great significance and can serve as a key link in the defense response. We saw continued upregulation of energy (C) and membrane biogenesis (M) genes and fewer DEGs in category F at 48 h vs. 24 h, which aligns with compromised nutrient processing as the midgut is destroyed.

### 3.7. Functional Enrichment Analysis of Differentially Expressed Gene GO

*S. marcescens* treatment induced dynamic changes in insect gene expression (Figure 13). At 24 h post-treatment, most genes were associated with metabolic and cellular processes, and many DEGs were involved in basic cellular metabolism and binding functions. By 48 h, the number of DEGs in these categories increased, suggesting an amplified host response. Notably, genes related to catalytic activity were more numerous at 48 h, possibly reflecting boosted metabolic changes in the infected gut. Pathogenic interaction triggered *S. marcescens*-mediated upregulation of genes governing metabolic and catalytic functions, with concurrent increases in genes maintaining cell integrity and regulating signaling pathways.

### 3.8. Enrichment Analysis of the KEGG Pathway of Differentially Expressed Genes

KEGG pathway analysis (Figure 14) revealed significant differential expression between treatment and control groups at 24 h. At 24 h, upregulated pathways included lipid degradation and some repair-related processes (hemocyte proliferation, plasmatocyte differentiation). By 48 h, many more metabolic pathways were up, including amino sugar metabolism, fatty acid metabolism, and hormone biosynthesis. Additional metabolic pathways were activated, including fatty acid elongation, unsaturated fatty acid biosynthesis, and degradation of lysine, leucine, and isoleucine. Immune-related genes (e.g., lysosomal enzymes) exhibited higher expression versus both 24 h and controls. The Toll and Imd signaling pathways displayed time-dependent dynamic changes—continuously rising. Notably, autophagy-related genes were upregulated at 48 h, consistent with extensive tissue damage caused by *S. marcescens* ZJ9 infection. Overall, KEGG analysis shows enhanced catabolism (fatty acid oxidation, amino acid degradation) and stress-response pathways in infected larvae, while immune pathways were variably regulated.

### 3.9. The qRT-PCR Verification

qRT-PCR validation of 10 DEGs confirmed the RNA-seq trends. For example, LOC118272367 and LOC118274099 were strongly upregulated at 24 h post-infection (T-1 vs. CK-1), consistent with sequencing data. Conversely, some genes showed downregulation at 48 h in the treated group (e.g., LOC118274192, LOC118274410). In general, there was good agreement between RNA-seq and qRT-PCR fold changes for these targets (Figure 15).

## 4. Discussion

### 4.1. Pathogenicity of S. marcescens ZJ9 and Its Infection Route in FAW

We found *S. marcescens* strain ZJ9 to be highly virulent to FAW, consistent with its known insecticidal activity. Egg, larval, and pupal mortality increased with bacterial dose (Figure 1). Previous work has shown *S. marcescens* enzymes can degrade insect cuticle and gut membranes [11]; our histology confirms extensive midgut damage (Figure 3). Meanwhile, the phage-mediated depletion of *S. marcescens* has been shown to enhance the proliferation of beneficial bacteria. This suggests that *S. marcescens* may possess toxic properties in FAW, ultimately leading to insect mortality. To investigate this hypothesis, we conducted experiments to assess the toxic effects of *S. marcescens* on FAW, which confirmed our initial conjecture and established its critical role in FAW poisoning. Furthermore, our results from egg stage treatment demonstrated a positive correlation between the *S. marcescens* concentration and lethality towards FAW eggs, indicating that increasing fluid concentration can effectively elevate egg, pupal, and larvae lethality in FAW [23,24]. Lee et al. have identified a potent virulence factor produced by *S. marcescens*, which suppresses cellular immunity by degrading adhesion molecules, thereby contributing to bacterial pathogenesis [25]. These findings offer insight into the insecticidal activity of *S. marcescens* in the midgut of insect hosts, as well as their molecular interactions in the gut.

We confirmed *S. marcescens* in the hemolymph of infected larvae (not found in controls), satisfying a criterion of pathogen causality. Based on the observed symptoms, bacteremia developed following FAW infection with *S. marcescens* [26]. After infection by entomopathogenic fungi, the gut microbiome of FAW undergoes reconstruction, potentially impacting the insect’s immune response. In addition, other studies have found that intestinal flora also has an important impact on the reproduction of FAW [27]. Hence, it is reasonable to infer that this is probably indicative of a gastrointestinal infection. The intestinal barrier refers to the protective layer of the midgut of insects to prevent harmful substances and pathogens from entering the body cavity. When insects consume food tainted with bacteria, the intestinal barrier presents impediments [28]. Bacteria may cross the intestinal barrier into the hemolymph [29].

### 4.2. S. marcescens Infection Elicits Hemocytic Immune Response and Functional Impairment in FAW

Blood samples were collected at 24, 48, and 72 h post-treatment to assess blood cell count and morphology. A significant proportion of granulocytes (GRs) exhibited pseudopodia extension following infection with *S. marcescens*. Plasmatocytes (PLs) and GRs, pivotal components in cellular immunity, displayed notable alterations upon the onset of infection. The three common blood cells of FAW all exhibited a trend of proliferation. According to Strand et al., the substantial energy expenditure required for the production and maintenance of large numbers of blood cells in FAW may have implications for their growth and reproduction [30,31]. Therefore, the surge in plasmatocytes (Figure 2) indicates a significant consumption of energy, suggesting that their growth has been affected. It has been reported that PLs play a role in the formation of connective tissue. Experimental data shows a significant increase in the number of PLs in the control treatment, with no significant difference between 24 h and 48 h. It is speculated that despite major losses, the number remains at normal levels. However, in the FAW treatment group, the cuticle of PLs became thin and showed hemolymph redness due to bacteremia, indicating insufficient support for normal growth and structural development. Sphrulocytes (SPs) are mainly involved in transport and cuticle-related functions containing chitin important for insect cuticle formation [32]. However, chitinase, a secondary metabolite of *S. marcescens*, decreases the efficiency of SPs. Since *S. marcescens* produces chitinase, it could impair spherulocyte function (they are involved in cuticle formation) [33]. Furthermore, changes in the morphology of hemocytes involved in metabolite storage and immune response were observed; GRs may indicate migration to the damaged gut site for pathogen consumption. GRs are the main phagocytic cells, and their increased numbers at 48 h suggest recruitment to the infection site. The earlier fluctuation in GRs could reflect initial consumption of pathogens or modulation by bacterial factors.

### 4.3. S. marcescens Disrupts the Midgut Physical Barrier and Induces Microbiota Dysbiosis in FAW

The midgut peritrophic membrane (PM) usually protects gut cells. In treated FAW, the PM was disrupted by 24–48 h, and the epithelial layer became highly disorganized (Figure 4). This likely allowed *S. marcescens* to breach the gut barrier. We hypothesize that *S. marcescens* chitinases contributed to PM degradation, consistent with the observed collapse of midgut structure. Additionally, there was evidence of loss of the peritrophic matrix around the midgut region’s esophagus, thickening of the intestinal wall, and contraction of the intestine. In fact, a smaller gut likely impairs digestion [34]. The altered gut morphology would reduce feeding efficiency [35]. The main function of the peritrophic matrix is to isolate food from midgut cells, protect vulnerable cells from damage, and provide a barrier against pathogenic microorganisms [36]. The reduced secretory capacity of the peritrophic matrix makes insects more susceptible to invasion by pathogens, increasing their mortality. At 48 h, irregular gaps appeared in the midgut tissue, suggesting that *S. marcescens* may escape from the gut into the blood lumen. Since the PM isolates food from gut cells [37], its loss would expose epithelium to further damage and infection. Gerald et al. suggested that vacuolization of the midgut epithelium is due to the destruction of goblet cell microvilli, and globule cells extruded from diseased midgut epithelial cells are rarely observed in healthy tissue [38]. By 72 h, we observed numerous cells with large vacuoles, indicative of stress or degeneration. Adjacent tissue showed many new small cells, suggesting active regeneration. These morphological changes imply significant energy investment in midgut repair [39].

*S. marcescens* is a bacterium commonly found in soil and water sources, and it is also a common pathogen of insects. It is also found in some insect intestinal symbiotic microflora. *S. marcescens* grows in a wide range of environments, including plant endophyte communities and the animal digestive tract. Although *S. marcescens* is generally considered to be low or avirulent, it is still an opportunistic pathogen [40]. Based on the observation that Enterococcus had the highest contribution value in the control group, it is highly consistent with the result characteristics of LEfSe analysis, and thus it is highly likely to come from LEfSe analysis. This phenomenon indicates that Enterococcus plays a role in maintaining the homeostasis of intestinal microorganisms in FAW. After infection, Serratia itself dominated the gut and likely outcompeted or disrupted other symbionts. For example, Enterococcus, common in healthy FAW guts, was displaced. Some minor taxa (‘Others’ in Figure 6) expanded, possibly representing environmental bacteria that proliferate under dysbiosis.” This avoids an unsubstantiated “symbiotic relationship”. This could mean that these microbes are able to gain a competitive advantage under the influence of *S. marcescens* ZJ9 or that they have some symbiotic or interactive relationship with *S. marcescens* ZJ9 [41].

The midgut microbial community consists of a variety of microorganisms that maintain homeostasis through the stable environment, nutrient competition, and metabolites. Insect gut microbes have received much attention for their contribution to host life characteristics [42]. Some of these microorganisms protect the insect host from various adverse threats, such as pathogen infection and parasitic wasp infestation [43,44]. However, this stability is dynamically balanced and fluctuates with insect physiological changes and food intake. The host will also fine-tune its immune system according to changes in the gut microbiota to maintain a stable microbial state [45]. Cluster analysis showed that the relative abundance of *Serratia* increased significantly after treatment. Additionally, the proportion of “Others” has increased significantly, which is relatively low; it is speculated that “Others” may contain a large number of environmental microorganisms. These results suggest that *S. marcescens* infection disrupts the normal control of the gut microbiome, possibly allowing opportunistic bacteria to flourish [46].

A decrease in microbial diversity can lead to an imbalance in the gut flora, which can affect host health. For example, after Beauveria infection with FAW, the gut microbiome was rebuilt, with a significant increase in the relative abundance of *Proteobacteria* and *Serrella*. This suggests that changes in the gut microbiome may be related to the viability of the host pest. The introduction of *S. marcescens* significantly influenced the trend of microbial diversity in the midgut, leading to a disruption of equilibrium and a change in its composition. This indicates that the midgut system gradually loses its ability to regulate the structure and composition of bacterial contents, allowing unrestrained proliferation of environmental microorganisms from food sources.

A disrupted microbiota may weaken gut immunity and nutrient absorption [47,48]. In some insects, gut symbionts can protect against pathogens, whereas in others they can enhance susceptibility [49,50]. In our case, the bloom of *Serratia* and loss of Enterococcus may have undermined FAW gut defenses. These results suggest that the gut microbiota plays a crucial role in the insect immune system but may have different effects under different conditions.

The significantly increased microbial community in the control group may consist of intrinsic members of the FAW midgut microbial community that maintained a healthy balance under natural conditions in the absence of ZJ9 treatment. The genus Enterococcus is commonly found in the insect midgut and functions as gut colonizers [51], and the reduction in Enterococaceae may indicate inhibition of these microorganisms or loss of their competitiveness after ZJ9 treatment. Overall, the significant increase in specific microorganisms in the treatment group may be directly related to the presence and activity of *S. marcescens* ZJ9, leading to significant changes in the intestinal microenvironment that may alter pH, oxygen levels, and nutrient availability, thus affecting other bacteria’s growth. Since the tissue sections showed severe intestinal damage and possible rupture, the competitive exclusion was suspected, with *S. marcescens* possibly having a competitive advantage in resource competition [52], leading to the exclusion of other microbial communities.

### 4.4. Transcriptomic Impacts of S. marcescens ZJ9 on Metabolism and Immunity in the FAW Midgut

The microbial balance in the insect gut is essential for digestion and nutrient absorption in terms of physiological effects; an imbalance in the microbial community may affect these functions, thereby changing the intestinal microenvironment and increasing the risk of structural and functional damage to the intestinal wall [53]. Under stress by *S. marcescens*, transcriptome data indicate that infection triggered major metabolic shifts. Both the treated and control groups at 24 h and 48 h show higher expression of lipid metabolism genes, but the treatment group in particular showed marked changes relative to the control. Concurrently, genes for cell division, synthesis, and tissue repair were upregulated (Figure 13). This suggests FAW larvae mobilize energy reserves and activate repair processes in response to gut damage and substantial energetic cost [54].

The transcriptomic results showed that a large number of genes exhibited significant changes in expression levels after treatment with *S. marcescens*. Specifically, the treatment groups displayed varying degrees of gene expression changes at both 24 h and 48 h, involving various biological processes such as metabolic processes, immune response, and cell cycle regulation. GO functional annotation of differentially expressed genes and KEGG pathway enrichment analysis revealed possible mechanisms underlying the effects of *S. marcescens* ZJ9 treatment on FAW midgut tissue [55,56]. The results of GO analysis showed that the differentially expressed genes were mainly involved in metabolic processes, cellular processes, and molecular functions, especially the genes related to energy conversion, amino acid metabolism, and the stress response of the cells, which showed obvious changes [57]. KEGG pathway analysis further revealed that pathways related to FAW metabolism, such as carbohydrate metabolism and amino acid metabolism, as well as immune-related signaling pathways like those related to bacterial infection, were significantly affected. The results of these transcriptome analyses suggest that *S. marcescens* ZJ9 treatment affected the basic physiological metabolic function of FAW midgut tissue, especially in energy metabolism and immune response [58]. Some reports indicate that the composition of phenoloxidase and antimicrobial peptides was not increased after *S. marcescens* infection, indicating that immune evasion of *S. marcescens* occurs in honey bees [59]. Several antimicrobial peptide genes were unexpectedly downregulated at 48 h, which could indicate immune suppression by *S. marcescens* [60]. When insects are invaded by exogenous pathogens such as pathogenic fungi, phagocytic blood cells either engulf the pathogens or synthesize antimicrobial peptides to remove them, accompanied by melanonylation reactions [61]. Notably, genes in the serotonin biosynthesis pathway were upregulated, which is intriguing because serotonin can be involved in tissue repair and stress responses [62]. This could relate to the observed histological repair attempts (Figure 4, 72 h). This finding aligns with observed changes in tissue tears and hemolymph post-infection.

### 4.5. The Application Potential of S. marcescens in Suppressing FAW Growth by Disrupting Gut Microecological Balance

Overall, our study suggests that *S. marcescens* affects the gut microbiota of FAW in multiple ways. The invasion of *S. marcescens* into FAW will undoubtedly cause an imbalance of the intestinal microbiota, leading to impaired immune system and intestinal nutrient absorption function. Overall, this may result in delayed growth and poor development in insects. This study highlights the important potential of *S. marcescens* to control FAW in future pesticide design.

## 5. Conclusions

The primary objective of this study is to elucidate the lethal mechanism of *S. marcescens* against the FAW at the molecular and cellular levels. Through bioassays, midgut histopathological analysis, intestinal microbiota profiling, and transcriptomic analysis, we found that *S. marcescens* disrupts the midgut structure of the FAW, thereby adversely affecting its metabolic and immune functions. This disruption ultimately leads to midgut rupture, systemic sepsis, and insect mortality. Our findings may contribute to the development of potential biological agents for the prevention and control of FAW infestations. These findings are based on laboratory assays; the field efficacy and safety of using *S. marcescens* as a biocontrol agent remain to be tested.

## Figures and Tables

**Figure 1 insects-16-00933-f001:**
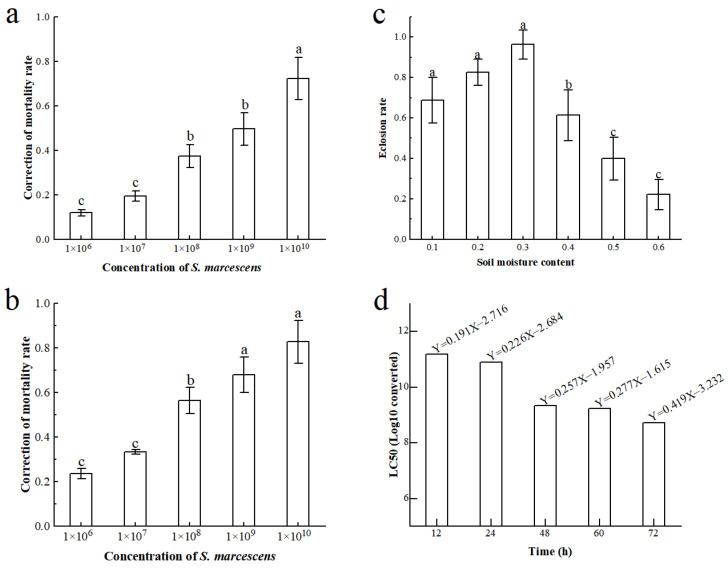
Toxic effects of *S. marcescens* on FAW. (**a**) egg mortality vs. *S. marcescens* concentration (horizontal coordinates are concentrations expressed in scientific notation, same below); (**b**) pupal mortality vs. concentration; (**c**) pupal eclosion vs. soil moisture; (**d**) larval LC50 curves. Different letters (a, b, c) above each bar denote significant differences (*p* < 0.05).

**Figure 2 insects-16-00933-f002:**
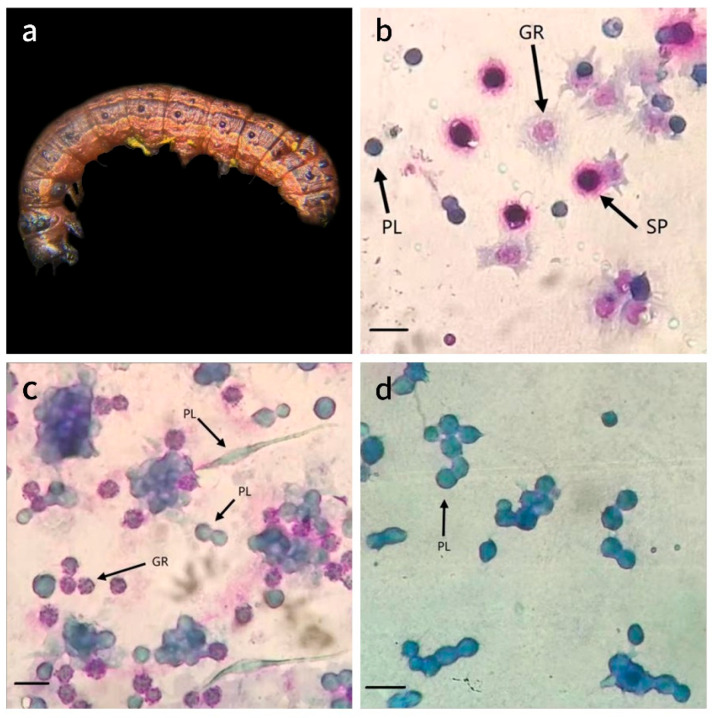
Observation of FAW larvae after infection with *S. marcescens* both in vivo and in vitro. (**a**) Characterization of FAW dying after 72 h treatment with *S. marcescens*. (**b**) Blood cells of the control group on the third day after infection. (**c**) Blood cells of the experimental group on the third day after infection. (**d**) Plasma blood cells showed agglutination.

**Figure 3 insects-16-00933-f003:**
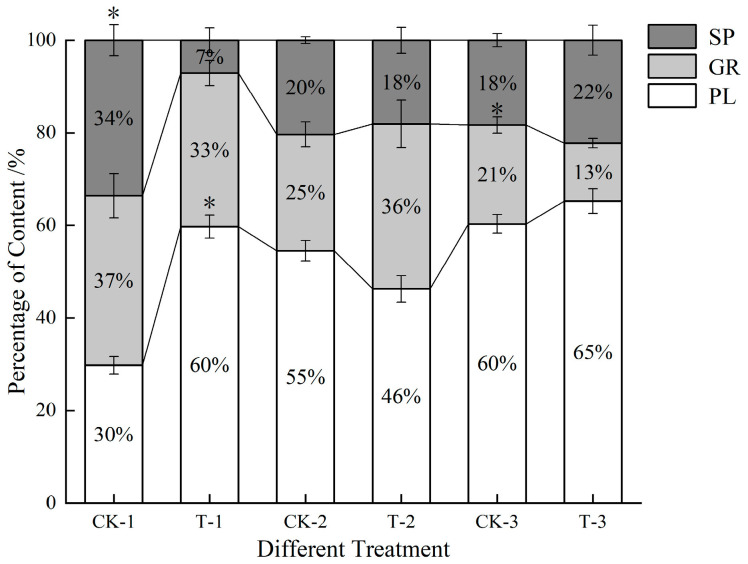
Dynamics of three types of blood cells of FAW larvae after *S. marcescens* infection. Note: CK-1 and T-1, respectively, represent the control group and the treatment group at 24 h, while CK-2, T-2, and CK-3, T-3, respectively, represent 48 h and 72 h. A column marked with “*” indicates a significant difference at the *p* < 0.05 level. (“CK = Control, T = Treatment”).

**Figure 4 insects-16-00933-f004:**
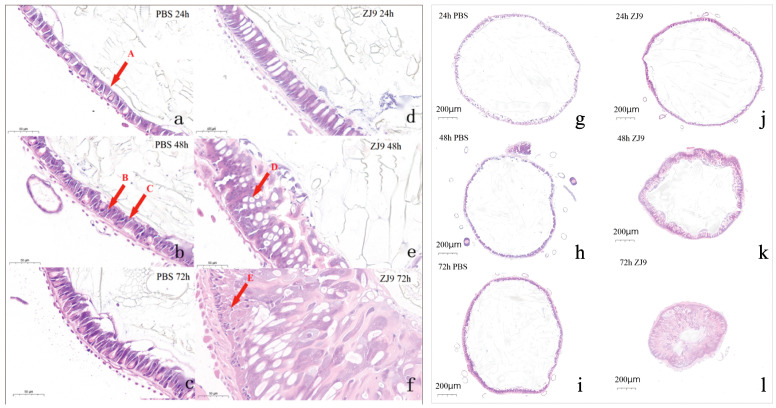
Middle intestinal tissue section. Note: (**a**–**f**) Local sections of FAW midgut tissue infected with *S. marcescens*. A: Peritrophic membrane (PM), B: goblet cells, C: columnar epithelial cells, D: vesicles, E: regenerating cells. The left scale is 50 μm. (**a**–**c**) are control treatments; (**d**–**f**) are *S. marcescens* ZJ9 treatments. (**g**–**l**) Cross-sectional changes in midgut tissue of FAW. (**g**–**i**) are control treatments; (**j**–**l**) are *S. marcescens* ZJ9 treatments.

**Figure 5 insects-16-00933-f005:**
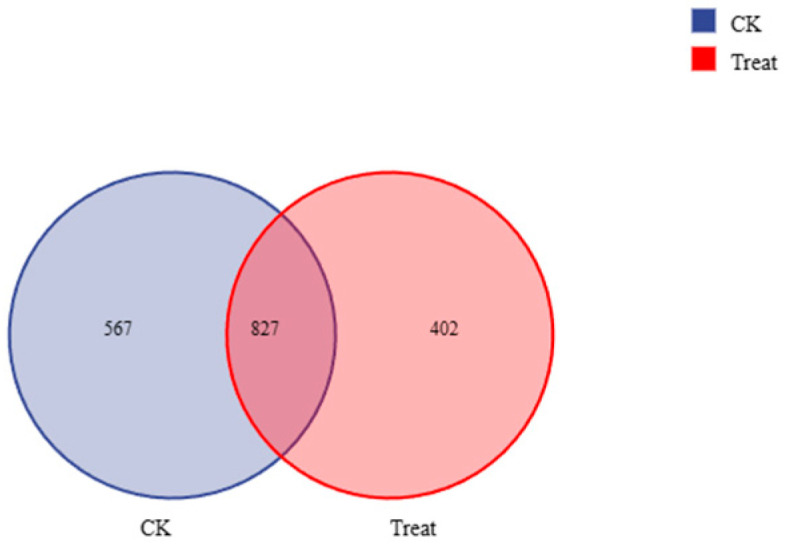
Venn diagram of species clustering control (blue) vs. treatment (red). Species clustering analysis identified 1394 bacterial species in the control group and 1229 in the treatment group, with 827 species shared between them.

**Figure 6 insects-16-00933-f006:**
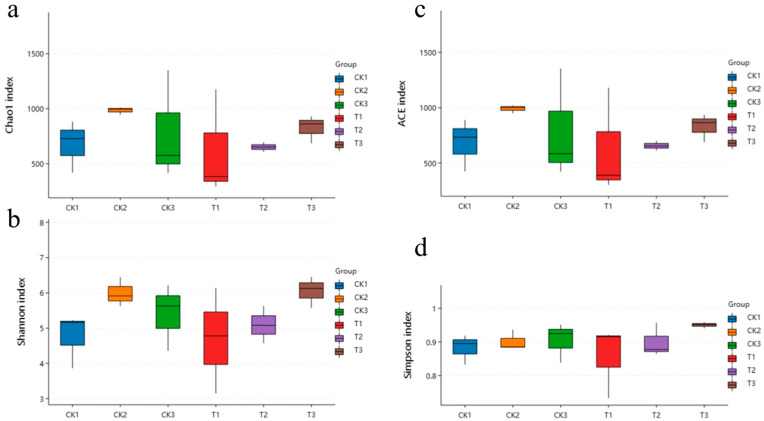
Effect of *S. marcescens* ZJ9 treatment on the microbial alpha diversity index of midgut tissue. Note: (**a**) Chao1 index (*y*-axes are Chao1 index), (**b**) Shannon index (*y*-axes are Shannon index), (**c**) abundance-based coverage estimator index (the *y*-axes are the abundance-based coverage estimator index), (**d**) Simpson index (*y*-axes are Simpson index), and CK1–3 and T1–3, respectively, represent the control group and the treatment group at 24, 48, and 72 h.

**Figure 7 insects-16-00933-f007:**
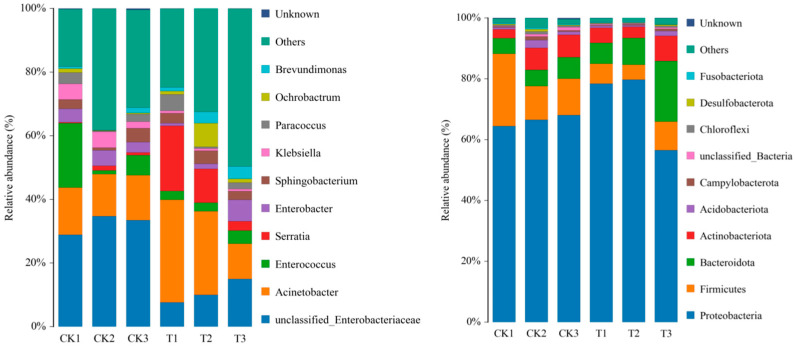
The changes in the relative abundance of intestinal microbiota between the treatment group (T1–3) and the control group (CK1–3) in Otus and genera (*y*-axes are relative abundance).

**Figure 8 insects-16-00933-f008:**
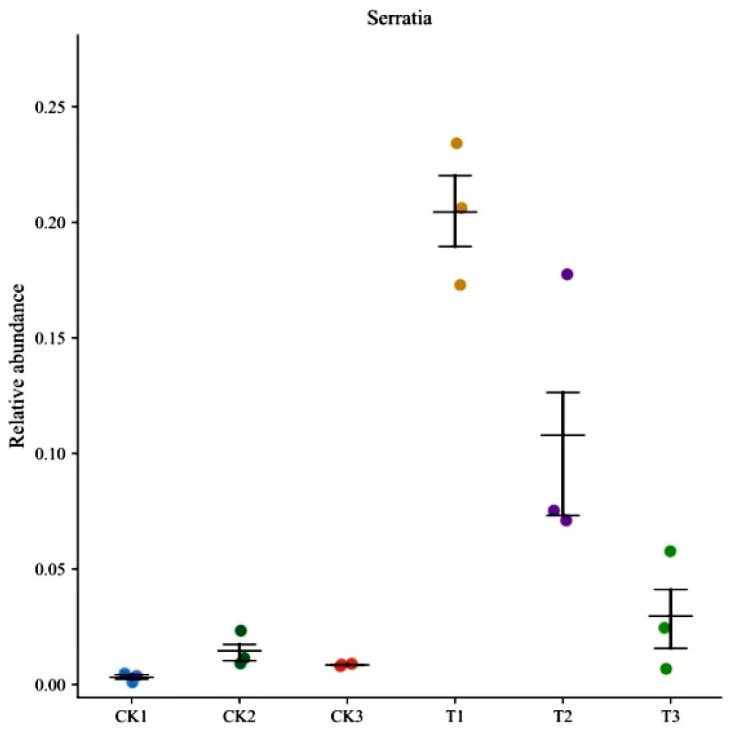
Changes in the control group (CK1–3) vs. treatment group (T1–3) in the relative abundance of *Serratia* spp. in the FAW midgut (*y*-axes are relative abundance).

**Figure 9 insects-16-00933-f009:**
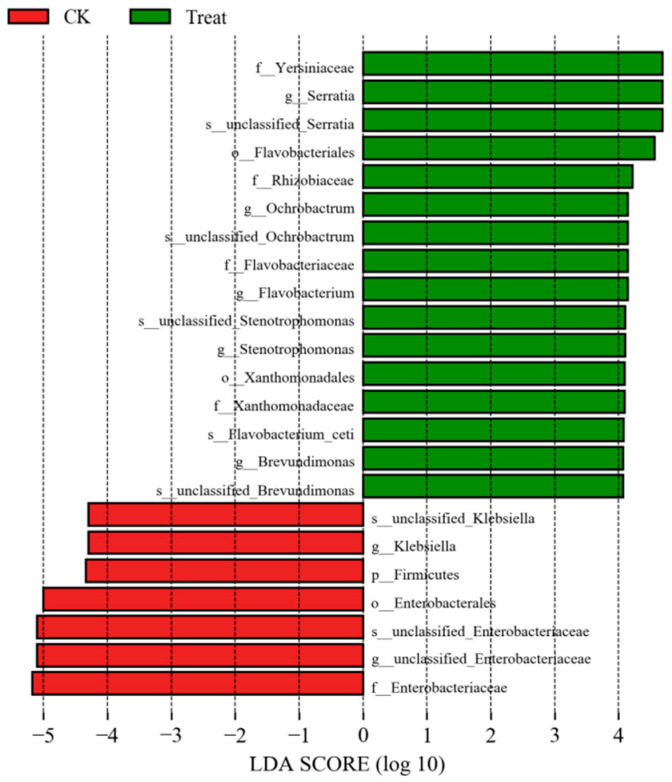
LEfSe (LDA) analysis identifying taxa enriched in control (green) vs. treatment (red) (the *x*-axis is the LDA SCORE).

**Figure 10 insects-16-00933-f010:**
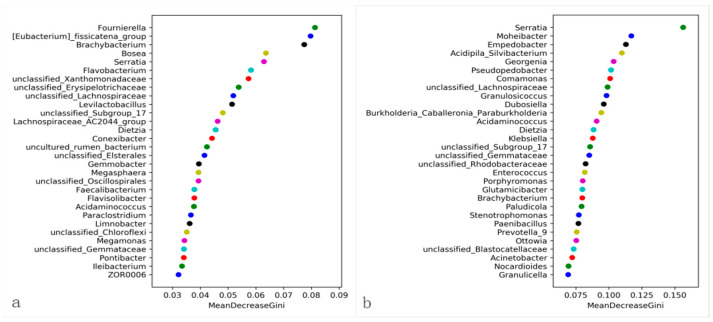
The random forest distribution between the control group (**a**) and the treatment group (**b**) (the *x*-axes are the mean decrease Gini).

**Figure 11 insects-16-00933-f011:**
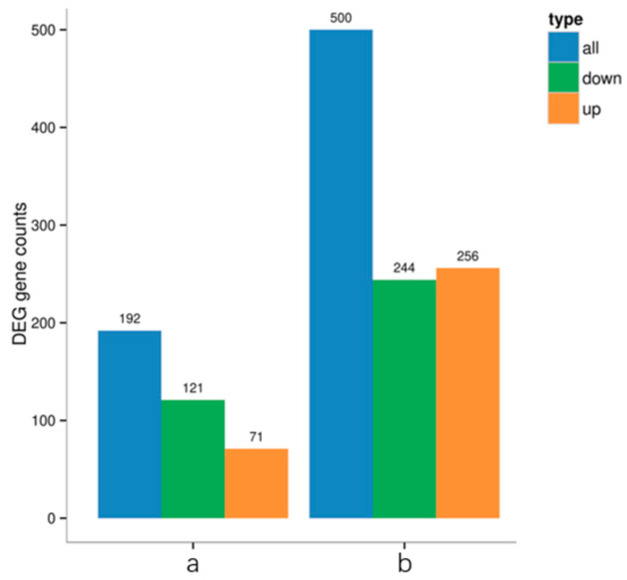
Number of differential gene changes. Note: a and b, respectively, for the 24 h and 48 h treatment groups and the control group.

**Figure 12 insects-16-00933-f012:**
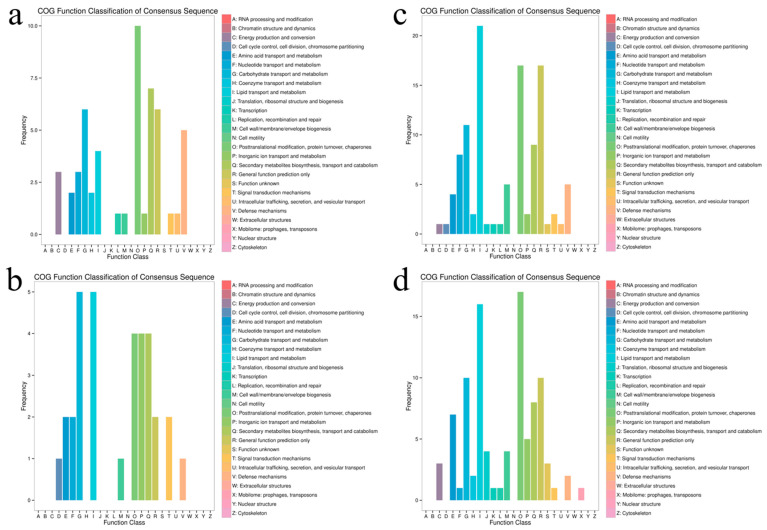
COG functional annotation of 24 h and 48 h differential genes. Note: (**a**,**b**) represents 24-h processing, and (**c**,**d**) represents 48-h processing. The upper bar shows the up-regulated genes in the treatment compared with the control. Lower bars show down-regulated genes in the treatment compared to the control.

**Figure 13 insects-16-00933-f013:**
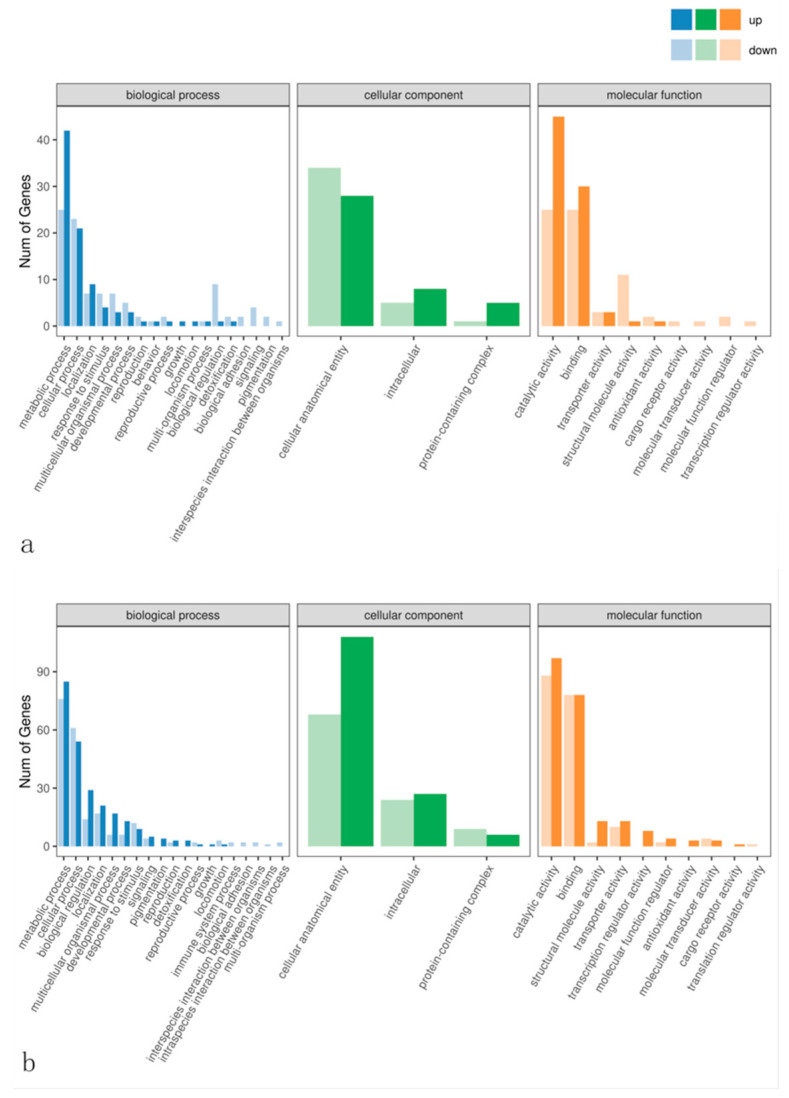
Annotation of GO functional cellular components, molecular functions, and biological processes of differential genes. Note: (**a**) and (**b**), respectively, for the 24 h and 48 h treatment groups and the control group.

**Figure 14 insects-16-00933-f014:**
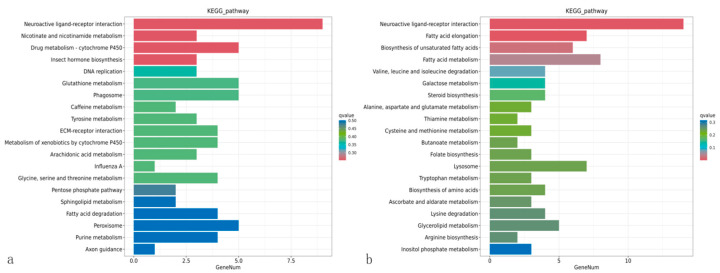
KEGG enrichment analysis of differential genes. Note: (**a**) and (**b**), respectively, for the 24 h and 48 h treatment groups and the control group.

**Figure 15 insects-16-00933-f015:**
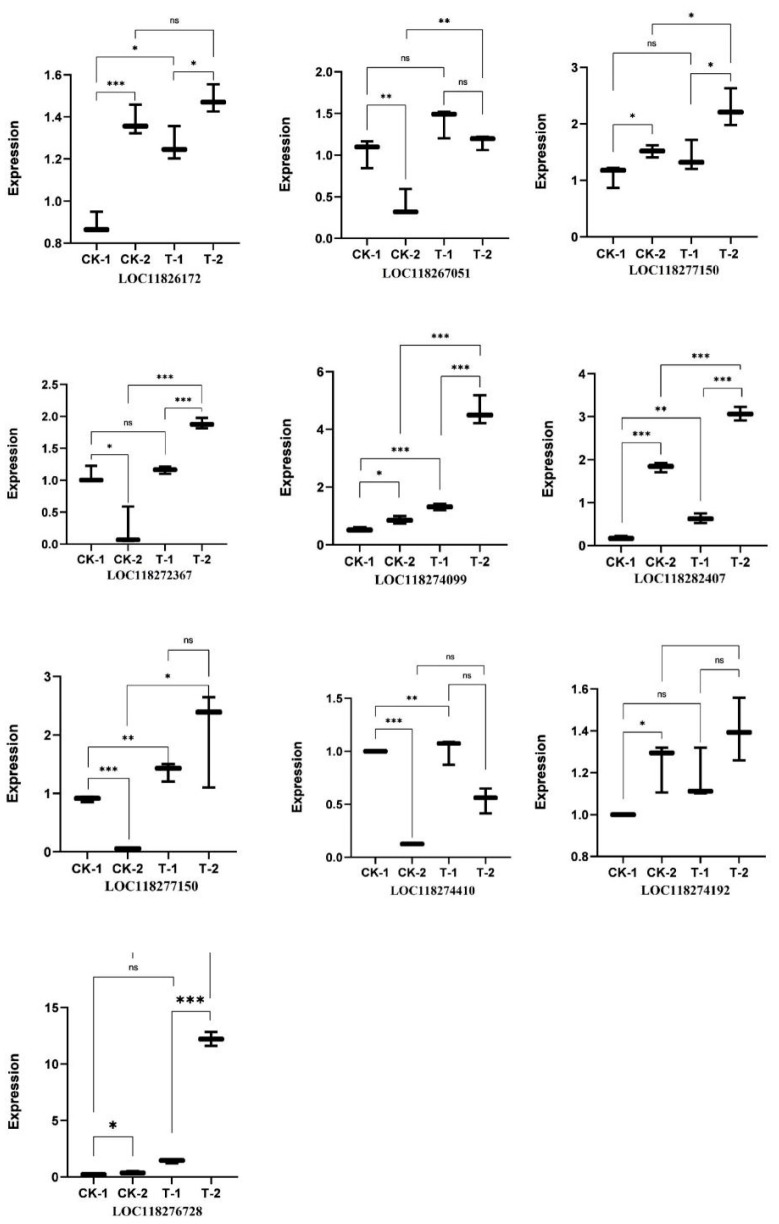
The qRT-PCR verification results. Note: CK-1 and CK-2 respectively represent the control groups at 24 h and 48 h; T-1 and T-2 represent the treatment groups for 24 h and 48 h, respectively. In the figure, the lowercase letter “ns” denotes a statistically insignificant difference (*p* > 0.05); a single asterisk (*) indicates a significant difference at the *p* < 0.05 level, double asterisks (**) indicate a significant difference at the *p* < 0.01 level, and triple asterisks (***) indicate a significant difference at the *p* < 0.001 level.

**Table 1 insects-16-00933-t001:** Primers utilized in the qRT-PCR experiment.

Gene Number	Upstream Primer	Downstream Primer
LOC118261729	CGTCGCCAAAGTGAAAGTGG	TTCGCATCGATGTAGGGCTC
LOC118267051	TAGCAGCCAGAGGTTTCCTG	AACAAGCCTTGAGTACCGCA
LOC118272367	GACTAGGTCACGCTGGCATT	AGACCGCTTGTGAACGTGAT
LOC118274099	CCCACCATCAACCCCAGATT	CAGCGAAACCACTGAGGACT
LOC118274192	AGAAGAGAACTCGGACGGGA	AGCACACCTGGTTAGCGAAA
LOC118274410	CGCATTGTCACAAAATGACACC	AGTGATGTCGGCTTTGCCTA
LOC118276728	AGTACAGCGTGAACCAAGGG	TTAACTTCGACGGCAACGGA
LOC118277150	ACGTTGCAGCTGCTGATTTG	TGTGCTACGGGAGAGTTTGC
LOC118282336	CAGTATCTGGCGTCCTCGTC	CACCGAGAGTTGTCTCGCAA
LOC118282407	AGCGATCATGAAATTGTTGGTGT	ACCTGTGCCGTCACTTTGAT
LOC118279579	TTGGTAGGCACGCTACACAG	CGGTGTCAGGCAGAAGATGT

## Data Availability

Due to privacy regulations and the large volume of data, the original data have not been included in this article.

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
