# Peer review of "Changes in Gut Microbiota, Midgut Structure, and Gene Expression of Spodoptera frugiperda Infected by Serratia marcescens"

_insects, 2025, doi:10.3390/insects16090933_

Round 1
Reviewer 1 Report
Comments and Suggestions for Authors
How was the microbiota analysis performed? Please provide more details on the microbiota determination, for example, how was the DNA purified? What technology was used? What bioinformatics tools were used for its analysis? Citations are required in this section.
For the transcriptome, please provide more details on the methodology. How was the RNA purified? What technology was used in transcriptome sequencing? What bioinformatics tools were employed? Citations are required.
The description in Table 1 does not provide enough information for its understanding.
How was the LC50 mentioned in the results calculated? This information is not included in the Materials and Methods section.
The figures need improvement because they look blurry.
In Figures 12-13, some panels are incomplete. Divide these figures into panels a, b, c, and d, and describe them as a single figure.
The scientific names on line 63, in Figure 8, and on line 302 should be italicized.
There is no mention of how the KEGG analysis was performed.
The numbering of Figures 16 and 17 is incorrect; they should be 15 and 16.
Were the genomic data deposited in a database? Include the accession numbers.
Author Response
Thank you very much for taking the time out of your busy schedule to review our manuscript. We have responded to all your suggestions for revision one by one. We hope that the revised manuscript will meet the standards for journal submission. We have placed the reply letter in the attachment in the form of a word document. Please check it when you have time.

Reviewer 2 Report
Comments and Suggestions for Authors
Summary
This manuscript investigates the lethal effects of the bacterium Serratia marcescens (strain ZJ9) on the fall armyworm (Spodoptera frugiperda) by examining changes in midgut histology, hemolymph (blood) cell composition, gut microbiota, and host gene expression. The authors report that feeding FAW larvae a suspension of S. marcescens causes midgut damage (peritrophic membrane rupture, narrowed gut lumen, cellular disorganization), systemic infection (bacteremia evidenced by isolating S. marcescens from hemolymph), and ultimately increased mortality at the egg, larval, and pupal stages. They also show altered hemocyte (blood cell) counts (a transient surge in plasmatocytes followed by fluctuations in granulocytes and spherulocytes), a disrupted gut bacterial community (marked increase in Serratia spp. and other taxa, higher alpha diversity by 72 h), and broad changes in the midgut transcriptome (upregulation of metabolic and tissue-repair genes, downregulation of lipid synthesis and some immune/hormonal genes). The authors conclude that S. marcescens impairs gut integrity and host immunity, leading to sepsis and death, and suggest this bacterium could be a potential biocontrol agent against FAW.
The study is interesting and addresses a significant pest problem. The multi-level approach (histology, microbiome, transcriptomics) provides depth. However, there are important issues to address before publication, related to clarity, experimental detail, data analysis, and interpretation. Major concerns include several inconsistencies and omissions in methods, unclear figure captions, and some overstatement of results. Additionally, the manuscript has numerous grammatical errors and formatting issues. Below I provide detailed feedback on required revisions.
Major Revisions
- Title and Species Names: The title contains a stray “3” before “transcription levels” (line 2–3) and should be corrected to “transcription levels” or “transcriptome”. The species name Spodoptera frugipeda is misspelled in the Simple Summary (line 17) and again in the Abstract (line 23); it should be S. frugiperda consistently. Ensure all scientific names (e.g. Serratia marcescens, Spodoptera frugiperda) are italicized and spelled correctly. After first mention, abbreviate to S. marcescens and FAW, respectively.
- Abstract and Simple Summary: Fix minor errors and improve clarity. For example, the abstract’s phrases “leading to systemic sepsis and subsequent mortality” and “value agent” are awkward. Clarify which life stages were tested and ensure that statements are supported by data. The Simple Summary repeats “mortality” twice; consider rephrasing to avoid redundancy. Also, explain abbreviations like FAW on first use.
- Introduction Context and References: The introduction generally covers FAW importance and S. marcescens traits. However, some claims need clearer references or rewording. For example, line 45–47 cites FAO global alert listing; verify this reference. The statement about 150 species of parasitic wasps (line 49) should cite a recent review. When discussing Serratia enzymes and effects on chitin (lines 60–67), be sure to connect directly to FAW context. The introduction could be streamlined: avoid overly detailed general info (e.g., lines 73–79 on gut microbes); focus on known interactions of Serratia and lepidopteran pests.
- Experimental Methods – Clarity and Reproducibility: Several methods lack detail or are scattered.
- Bioassays on Eggs and Pupae: The results refer to egg and pupal treatments (Fig. 1a–c), but the methods section does not fully describe how eggs and pupae were exposed to S. marcescens. For instance, lines 194–201 mention treating eggs and soil with bacteria, but no methods section covers these experiments explicitly. Please add a clear description of the experimental setup for egg and pupal assays (e.g., how eggs were inoculated, how pupae were reared in soil with bacteria, how mortality and eclosion were recorded, sample sizes, controls). The term “soil relative water content of 30%” (line 198) appears without context in methods; explain how pupae were handled and why soil moisture was varied.
- Microbiome Sequencing: The manuscript states that midguts were collected and microbial diversity analyzed using a “Biomarker microbiome analysis platform” (line 145). Please specify the sequencing method (e.g. 16S rRNA gene region, sequencing platform) and bioinformatics pipeline in the methods. Clarify how DNA was extracted, how many reads or OTUs were obtained, and what metrics (Shannon, Simpson, etc.) were used for alpha diversity. Also, describe how samples were randomized and if blank/negative controls were included.
- Transcriptome and qPCR: It is unclear why both DESeq2 and edgeR were used (line 160). Typically one method suffices; clarify whether both were applied and how results were reconciled. State the reference genome or transcriptome assembly used for mapping, and mention quality controls (e.g. read counts, replicates). For qRT-PCR (line 173–180), specify what “the attached table” refers to (the primers are in Table 2) and whether technical replicates were averaged. It may be clearer to merge sections 2.5 and 2.6. Also, the “fold change ≥2” criterion (line 162) should specify log2 fold-change or linear fold-change consistently.
- Data Analysis and Statistics: The manuscript often presents percentage changes or trends without stating statistical significance. For example, the changes in hemocyte counts (Fig. 3) are described in percentages relative to control, but it’s not clear if differences are statistically significant. Please report p-values or confidence intervals where appropriate, and describe the statistical tests used (ANOVA, t-test, etc.) including how normality/homogeneity assumptions were checked. In figure legends, include error bars and sample sizes. For the LC50 determination (Fig. 1d), describe how the regression was fitted and if it meets accuracy criteria (provide R² and confidence intervals). All major result statements should be backed by statistical analysis.
- Figure Presentation: Several figure captions and panel labels are confusing:
- Figure 1: The caption text (lines 207–213) merges panel descriptions with commentary. It’s better to split panels (1a–d) clearly: e.g., 1a = egg mortality vs. S. marcescens concentration; 1b = pupal eclosion vs. soil moisture; 1c = pupal mortality vs. concentration; 1d = larval LC50 curves. Ensure the caption specifies what each panel shows and how the data were obtained (e.g. error bars, replicates).
- Figure 4: The left and right images are labeled differently (left: “Local sections …”, right: “Cross-sectional changes …”). These should be integrated or labeled as sub-figures (4a–f). The caption’s notation (“A: Encircling film, B: Cup-shaped cells, …”) is not clearly matched to the images. It might be better to annotate structures in the images with arrows/labels and then describe them in the caption. Make sure the terminology is correct (e.g. “peritrophic membrane” instead of “encircling film” if appropriate). Also, clarify which panels correspond to control vs. treated (the caption mentions a–c vs. d–f, but the text of the manuscript should explicitly reference these).
- Figures 5–10: For microbiome figures (Venn, diversity, LEfSe, random forest), ensure each figure caption explains what is plotted (e.g. “Figure 9: LEfSe (LDA) analysis identifying taxa enriched in control (blue) vs. treatment (red)”). The random forest (Figure 10) caption says “a: Control group, b: Processing group” but the plotted “feature importance” needs context (e.g. top discriminative genera). It’s unusual to have separate RF models for two groups – likely it’s one model distinguishing control vs. treatment samples; clarify this.
- Figures 11–14: The gene expression figures (COG, GO, KEGG) have captions that refer to “upper bar” and “lower bars” etc. It would help to explicitly label “Up-regulated genes (treatment vs. control)” and “Down-regulated genes”. Also, ensure figure numbers match text references (e.g. the text refers to “Figure 15a, b” for KEGG but then “Figure 16” appears in the caption at line 379). Double-check numbering and panel labels throughout.
- qRT-PCR (Figure 17): The caption should clarify what CK-1, CK-2, T-1, T-2 represent (it is explained in text but should be in caption too). Indicate that error bars are standard deviation and note significance markers (if any).
Overall, the figures need clearer legends and consistency with the text. Consider moving some figure details (like notes a, b, c for CK/T groups) into the legend.
- Interpretation and Biocontrol Claim: The conclusion that S. marcescens “may serve as a valuable agent for controlling FAW” needs a cautious tone. S. marcescens is known to be an opportunistic pathogen (even in humans) and can harm beneficial insects; this ecological/safety aspect isn’t discussed. The authors should at least acknowledge potential risks or mention that future work must assess non-target effects and formulation. Suggest adding a sentence discussing the balance between efficacy and safety (and referencing literature on Serratia safety if available).
- Discussion Depth and Organization: The discussion covers many points but sometimes jumps between subjects. It would benefit from re-organization. For example, separate paragraphs could address: (1) confirmation of pathogenic effect (link hemocyte changes to mortality); (2) midgut damage and its consequences (connect histology to “sepsis” hypothesis clearly); (3) microbiome shifts (tie observed changes to functional implications); (4) transcriptomic changes (focus on key pathways rather than enumerating all); (5) implications and future work. The current text sometimes speculates without citing (e.g. line 572-577 on serotonin); ensure such claims are directly supported by the data or references. Also, some sentences in discussion are hard to follow (for instance, line 542–549 about metabolism genes), so revise for clarity. Avoid over-general statements (e.g. the broad statement that producing many hemocytes disrupts physiology [line 437–439] – if citing Strand et al. [32,33], ensure these refs actually support it).
- Language and Style: The manuscript has many grammatical errors and awkward phrases. For example, “plasmatocyte (PL) typically exhibited a round or oval shap” (line 222–223, “shape” typo), “plasmatocyte shap, with some displaying partial spindle shape.” or “cup-shaped cells” vs. “columnar cells” uses inconsistent vocabulary. A native English speaker should thoroughly edit the text. Correct terminology where needed (e.g. the “perifeeding matrix” mentioned in the discussion [line 454] should be “peritrophic matrix”). Ensure verbs agree and tenses are consistent. Spell-check (“shap” → “shape”, “prominently bead-like inclusions” vs. “bead-shaped”, etc.).
Address the following minor points in editing (see next section), but a full language review is recommended.
Minor Revisions
- Abstract and Keywords: In the Abstract, “Spodoptera frugipeda” should be frugiperda. In Keywords, “transcription level” might be better phrased as “transcriptome analysis” or “gene expression”.
- Consistency in Abbreviations: Define FAW (fall armyworm) on first use (you do on line 23–24, OK). Consider defining abbreviations like LB (Luria-Bertani) medium, PBS (though LB is common, PBS was expanded to phosphate-buffered saline, which is fine). Use “CFU/mL” uniformly (currently it’s “cfu/mL” at line 116 vs “CFU/mL” elsewhere). Use degrees symbol properly (28 °C, with a space) rather than “28℃”.
- Units and Spacing: Ensure a space between number and unit (e.g. “10 g/L”, “180 rpm”, “4 μm”, “95 ℃” should be “95 °C”). Check that each figure caption and text uses SI units correctly and consistently.
- References and Citations: The text includes numbered citations [e.g. 1, 2, 5,6, etc.]. Ensure the final version has a complete reference list. There are also numbered references within paragraphs (e.g. [19–21]) – verify these correspond to cited work. In the review, we do not have the reference list, but please check that all citations are present and formatted per journal guidelines.
- Figure Captions (minor):
- In Fig. 2 caption (lines 244–248), clarify that (PL) plasmatocyte, (GR) granulocyte, (SP) spherulocyte are indicated. The caption letters (a, b, c, d) should match the panel letters and images in the figure.
- Fig. 3 caption (lines 250–253) is mostly clear, but define “CK” and “T” in the legend if not already defined in text (“CK=Control, T=Treatment”).
- In Fig. 4 caption (lines 273–278), ensure “peritrophic membrane” is used uniformly rather than “encircling film” (if that’s what “A:” stands for).
- The Venn diagram (Fig. 5) caption should say what the two sets are (control vs treatment) and what the numbers represent (species, OTUs?).
- For Figs 6–10, clarify if y-axes are indices (e.g. Shannon diversity), relative abundance (%), LDA score, feature importance units, etc. The random forest importance (Fig. 10) is likely Gini importance or something; label the axis.
- In KEGG/GO figure captions (Figs 14–16), “Figure 16” is out of sequence; check numbering.
- Tables: Table 1 (lines 190–190) heading is “Sequencing data of randomly selected genes.” This is confusing because it looks like log2FC values from transcriptome comparisons, not “sequencing data”. Either rename it to “Fold changes for selected DEGs” or similar, or integrate this into text. Also, the gene IDs (LOC118... numbers) are not informative – if possible, give gene names or putative functions alongside (e.g. “LOC118276728 (FBXL4-like)” if known). Table 2 (lines 191–193) is fine but the column headings “Upstream primer” should be “Forward primer” and “Downstream primer” → “Reverse primer”.
- Gene Expression Validation: The narrative for qRT-PCR (lines 382–400) is hard to follow. It might be clearer to summarize as “We randomly selected 10 DEGs with |log2FC|>2 for qRT-PCR validation using RPL13 as reference. The qRT-PCR results (Fig. 17) confirmed the trends seen in RNA-seq: e.g. gene LOC118272367 and LOC118274099 were significantly upregulated at 24 h post-infection, while other genes showed downregulation or no change consistent with the transcriptome data.” In the figure legend, explain what CK-1, CK-2, T-1, T-2 mean (I assume CK/T = control/treatment, 1/2 = 24h/48h). Also, mark statistical significance on bars (e.g. * for p<0.05). Ensure consistency: in the text you call it “T-2 (day 2) vs controls”, but in fig legend use “48h” instead of “day 2”.
- Terminology: Standardize terms. For example, use “peritrophic membrane (PM)” consistently instead of alternate phrases. Be careful with terms like “bacteremia” vs “sepsis” – you show bacteria in hemolymph (bacteremia), sepsis implies systemic symptoms. It’s okay to say “systemic infection” if sepsis is too strong. Also, “non-infected” larvae in abstract vs “control” elsewhere – use one term (prefer “control” or “uninfected”).
- Units and Numbers: Check if “LC50=7.50×10^7 cfu/mL” is written with correct format (should be CFU/mL and use a space or superscript). When describing percent mortality or changes, ensure consistent rounding/precision and percent sign (e.g. “108.21% of control”).
- Editorial: Remove placeholders like “Received: date, Revised: date” etc. Ensure all headings and subheadings are correctly numbered (Methods has 2.x, Results 3.x, etc). The author listing and affiliations (lines 5–13) have numbering “1,2,3” etc; verify consistency per journal style (some may want superscript or symbols rather than plain numbers). Also check the contact email formatting (line 13).
Specific (Line-by-Line) Comments
- Line 3–4: The title reads “Changes in the gut microbiota, cellular structure, and 3 transcription levels of Spodoptera frugiperda…”. Remove the stray “3” and consider rephrasing “transcription levels” to “transcriptome”. For example: “Changes in gut microbiota, midgut structure, and gene expression of S. frugiperda…”.
- Line 17: In the Simple Summary, “Spodoptera frugipeda” is misspelled. Correct to frugiperda.
- Line 23: Abstract also uses “Spodoptera frugipeda”. Fix to frugiperda.
- Line 60–62: “the peritrophic membrane (PM), a protective covering in the digestive tracts of numerous insects [9,10].” Possibly explain why chitinases matter: S. marcescens chitinases could degrade the PM and gut lining, linking to pathogenicity.
- Line 70: “The midgut plays a crucial role... as a key site for food conversion.” Consider reorganizing – some info (e.g. Lepidoptera midgut is second largest organ [line 74]) might be streamlined.
- Line 81: The sentence beginning “The strain S. marcescens has been reported to have insecticidal activity against FAW [22].” is fine but perhaps move it earlier to directly tie into the study aim.
- Line 91–98: Minor copyediting: “The larvae were fed corn leaves from the Zhengdan 958 variety cultivated in the greenhouse…” is fine. Check hyphenation (“16L:8D” → “16 L:8 D” or “16:8 h light:dark”).
- Line 95: after “maintained at 28℃”, the “℃” should have a space (28 °C).
- Line 98–99: The description of strain ZJ9 is clear; perhaps clarify its origin (environmental isolate or from FAW gut?) if known.
- Line 111–117: The LB medium recipe and bacterial culture steps are very detailed. It may be more concise: some details (e.g. phrase “mother liquor”) can be simplified. Also ensure consistency: “LB agar plates” vs “LB solid medium” – stick to one term.
- “CFU counting by dilution plating” is fine, but specify how (serial dilutions on LB agar, colony count).
- Line 123–128: The method for hemolymph collection is described. Possibly note that anesthesia on ice is to slow movement. The cell counting (“about 300 cells counted per field”) – are counts averaged over fields? How were statistical comparisons done? This might be too detailed for a Methods section; consider summarizing key steps.
- Line 134–139: “Two hundred fourth-instar larvae of uniform growth were selected...” – Here it describes feeding experiment. Clarify: were larvae starved for 12h before feeding (line 137)? After feeding, were they given only treated leaves? It says leaves replaced daily for 3 days (lines 137-140). Good to state how many larvae per cup, though it says “each cup contained 8 leaf disks” but how many larvae per cup? (I think one per cup since they were isolated in cups). If multiple per cup, also mention replicates.
- Line 146: “Microbial abundance and diversity were analyzed using the Biomarker microbiome analysis platform.” – We need a bit more: e.g. was this QIIME-based? Did you use 16S sequencing? Cite a method or this platform’s website if possible.
- Line 149–154: Histology methods are mostly standard. You mention slides stained with H&E and observed by light microscope; consider adding magnification or microscope model if relevant.
- Line 156–164: For RNA-seq, clarify “each replicate” (i.e. how many biological replicates per group). The text suggests 3 reps. State them (e.g. “Three biological replicates were used for each condition”). Also, the description “differential grouping was performed using edgeR” is unclear – normally one would say “DEGs were identified with DESeq2 (or edgeR) using |log2FC|≥1 and FDR<0.01.” Pick one tool for consistency, or explain if you used both (for cross-validation?).
- Line 174–179: The qRT-PCR protocol is given in detail. Good to specify how you calculated relative expression (ΔΔCt method?). Mention that RPL13 was validated as a stable reference (as [23] presumably shows). It’s acceptable, but double-check that all reagents (e.g. “Evo M-MLV RT mix”) are correctly named and that thermocycling steps are clear (e.g. 95 °C for 30 s, 40 cycles).
- Line 190–191 (Table 1): The table title “Sequencing data of randomly selected genes” is confusing; these values look like log2 fold-changes. Consider retitling “Log2 fold-change of selected genes (treatment vs control)”. Also, column headers “T1 vs T2, CK1 vs CK2, CK1 vs T1, CK2 vs T2” should be defined in a footnote (I infer: T=Treatment, CK=Control, 1=24h, 2=48h). This table appears mid-methods, but it’s actually results data – consider moving it to Results or supplement.
- Line 194–201 (Figure 1 results): It’s good that lethality increases with concentration. However, the text says “optimal eclosion at 30% soil moisture (Fig. 1b)”; this sounds like an unrelated experiment on moisture. If soil moisture was tested, explain why (maybe S. marcescens activity depends on moisture?). If not central to the study, this part may be trimmed or explained better. The phrase “increased significantly after reaching 1×10^7 cfu/mL (LC50=7.50×10^7)” is confusing: LC50 is the concentration causing 50% mortality, not “reached after”. Rephrase: e.g. “We observed a significant rise in pupal mortality above 1×10^7 CFU/mL, with an estimated LC50 of 7.50×10^7 CFU/mL.” Also clarify which life stage the LC50 refers to (the caption says “fourth instar larvae at different time intervals” – was this a time-dependent LC50?).
- Line 215 (Figure 2 description): The description “characterization of FAW dying after 72h treatment” suggests Fig. 2a shows a diseased larva. Make sure the images in Fig. 2 correspond (they mention Fig. 2a and 2b). The caption “(The same blow)” seems like a typo. Also, the note “PL, GR, SP (The same blow).” is unclear. Perhaps the caption was cut off. Ensure caption only describes panels; the PL/GR/SP definitions should be in main text or figure legend clearly.
- Line 222–230 (Blood cell descriptions): There are some typos and needed clarifications:
- “plasmatocyte (PL) typically exhibited a round or oval shap” → “shape”.
- “often clustering together (Figure 2c). These cells were basophilic, showing a light blue coloration following staining.” – fine.
- “Granulocyte (GR) presented various shapes, predominantly round, with some extending pseudopodia. After staining, these cells contained heterogeneous purplish-red lysosomal granules (Figure 2b).” – good.
- “Spherulocyte (SP) featured prominent bead-like inclusions…” – okay, but not sure “bead-like inclusions” are on the cell periphery? Possibly clarify location (“at cell periphery”).
- “No significant morphological differences were observed between PL and SP following infection; however, GR in the control group were primarily circular while those in the treatment group showed a notable proportion extending pseudopodia.” → This sentence is confusingly phrased. Perhaps split: “In the treatment group, plasmatocytes and spherulocytes appeared similar to controls. However, many granulocytes in infected larvae extended pseudopodia (phagocytic projections), whereas control-group granulocytes remained mostly round.” And cite Fig. 2a,b properly.
- Line 233–239 (Blood cell dynamics): The description of changes could be clearer:
- PL: “increased to 201.5% of control at 24h, then dropped to 84.9% at 48h, and 108.2% at 72h.” Note if any of these differences are statistically significant (especially the first drop).
- GR: “peaked at 48h (141.9% of control) then decreased”.
- SP: “fell to 21.1% of control at 24h, then recovered gradually to near control levels by 72h.”
The sentence “Overall, following infection… PL increased rapidly, SP declined, and GR fluctuated later” (lines 241–242) is a good summary; perhaps put it in simpler terms. Also, label the y-axis (%) in Fig. 3 if using percent. - Line 254–263 (Midgut histology): The results describe progressive damage. A few points:
- “midgut shrinkage and rupture, PM degradation, radial cracks in intercellular matrix” – these phrases should match the visible histology. If possible, simplify: e.g., at 24h midgut epithelium begins to separate (gaps), goblet cells elongate; by 48h peritrophic membrane mostly gone, enlarged intercellular spaces, forming vesicles; by 72h the epithelium shows many regenerative cells and the lumen is highly constricted.
- Ensure consistent use: if “columnar cells” or “cup cells” are used (line 270 “A: Encircling film, B: Cup-shaped cells”), define them. Perhaps “Columnar epithelial cells” and “goblet cells” if that’s what cup cells are.
- The text mentions “the treatment group exhibited a significant reduction in intestinal diameter” (lines 269–272) – if this is shown in Fig. 4 (right), state how this was measured (image analysis?) and if quantitatively significant.
- Avoid overly dramatic language like “complete disappearance of the PM” (line 268) unless the images clearly show no membrane; perhaps say “peritrophic membrane largely disrupted by 72h.”
- Line 279–284 (Figure 5–6 discussion): The microbiome Venn (Fig. 5) and alpha diversity (Fig. 6) are mentioned. Clarify wording: “1394 species in control, 1229 in treatment, with 827 shared” – do you mean OTUs or predicted species? The statement “significant increase in microbial diversity in treatment” then “decreased at 24h and 48h but increased at 72h” is contradictory. Possibly what you meant is that overall, treatment samples had higher diversity by 72h. Rephrase to: “Alpha diversity (Shannon index) initially dropped in treated larvae (24–48h), but by 72h the gut community was significantly more diverse than controls (Fig. 6).” Check the data to confirm this pattern. Also, if any statistical test (e.g. ANOVA) was applied, mention it here or in figure legend.
- Line 285–291 (Genus-level changes): The text correctly notes Serratia spiked at 24h. Perhaps quantify: e.g. Serratia made up X% of sequences at 24h in treatment vs Y% in control. The mention of “Others” category rising (line 291) is interesting; explain what “Others” refers to (all genera <1% abundance summed). If many environmental taxa appear, it suggests breakdown of normal community structure. Consider briefly naming any specific genera that emerged if relevant (or mention that many low-abundance taxa expanded).
- Line 303–310 (LEfSe and Random Forest): The LEfSe result that Serratia is a biomarker in the treatment and Enterococcus in control is plausible. The text should clarify: “Enterococcus was significantly enriched in control guts, whereas Serratia dominated the treated group (Fig. 9).” Also, in the random forest (Fig. 10), if Serratia ranks fourth in overall importance, what are the top three? Mentioning them could add context (they may be other genera that differed). If Enterococcus ranks high in control, that’s consistent. Just ensure any commentary matches the figure data.
- Line 317–324 (Transcriptome DEGs): The increase in DEGs from 24h to 48h is interesting. Explicitly state numbers (e.g. “At 24h post-infection, we found X upregulated and Y downregulated genes; at 48h, these increased to X’ and Y’”). The text says total DEGs at 48h were 2.6× the 24h count, but providing the actual counts (even approximate) is helpful. Also clarify what |log2FC|≥1 means (is that 2-fold up/down).
- Line 328–337 (COG analysis): Instead of listing letters, you might say “Upregulated genes were enriched in energy production (COG category C), amino acid metabolism (E), transcription (K), and cell wall/membrane biogenesis (M) (Fig. 12). Downregulated genes were enriched in RNA processing (A), chromatin structure (B), and cell cycle (D).” For 48h (Fig. 13), summarize “We saw continued upregulation of energy (C) and membrane biogenesis (M) genes, but fewer genes in coenzyme (H) and nucleotide (F) metabolism categories, which aligns with compromised nutrient processing as midgut is destroyed.” The current text “decreased gene frequency” is unclear—better to say “fewer DEGs in category F at 48h vs 24h.” Also, the mention of upregulation of signal transduction (T) seems important; highlight it as defense response perhaps.
- Line 351–359 (GO terms): It’s stated that genes in metabolic and cellular processes increased at 48h and that catalytic activity terms were higher. Instead of describing all bars, focus on key terms: e.g. “At 24h, many DEGs were involved in basic cellular metabolism and binding functions (Fig. 14). By 48h, the number of DEGs in these categories increased, suggesting an amplified host response. Notably, genes related to catalytic activity were more numerous at 48h, possibly reflecting boosted metabolic changes in the infected gut.” If possible, mention specific enriched GO terms (e.g., “oxidoreductase activity” or “carbohydrate metabolism”) if the analysis revealed them.
- Line 363–372 (KEGG pathways): Reword to clarify: At 24h, upregulated pathways included lipid degradation and some repair-related processes (hemocyte proliferation, plasmatocyte differentiation). By 48h, many more metabolic pathways were up, including amino sugar metabolism, fatty acid metabolism, and hormone biosynthesis. The statement “immune-related genes (lysosomal enzymes)” is vague; specify if lysosome pathway genes are up. Also, “Toll and Imd pathways showed dynamic changes” – were they up or down? If certain immune genes were downregulated (e.g. AMPs), mention that. The mention of autophagy genes is good; consider linking it to visible cell damage (cells might be autophagically dying). The text is a bit list-like; try to synthesize: e.g. “Overall, KEGG analysis shows enhanced catabolism (fatty acid oxidation, amino acid degradation) and stress-response pathways in infected larvae, while immune pathways were variably regulated.” The Fig. 15 (or 16) caption should align (it says “enrichment analysis”, but the figure probably shows enriched pathways).
- Line 382–390 (qRT-PCR summary): The current paragraph listing each gene’s pattern by name is hard to follow. Instead, the text can say: “qRT-PCR validation of 10 DEGs confirmed the RNA-seq trends. For example, LOC118272367 and LOC118274099 were strongly upregulated at 24h post-infection (T-1 vs CK-1), consistent with sequencing data. Conversely, some genes showed downregulation at 48h in the treated group (e.g., LOC118274192, LOC118274410). In general, there was good agreement between RNA-seq and qRT-PCR fold-changes for these targets (Fig. 17).” Avoid absolute statements like “significant up/down” unless backed by statistics on qPCR. The figure 17 should have error bars and significance stars if appropriate.
- Line 404–411 (Discussion start): The opening sentence is confusing: “Some researchers have suggested that the growth of housefly larvae was inhibited after consuming S. marcescens, … providing essential nutrients or digestive processes [13,14].” This seems misplaced – the references [13,14] likely relate to microbial symbionts (from earlier). Delete or separate these ideas. Focus discussion: e.g. “We found S. marcescens strain ZJ9 to be highly virulent to FAW, consistent with its known insecticidal activity [22]. Egg, larval, and pupal mortality increased with bacterial dose (Fig. 1).” Then transition: “Previous work has shown S. marcescens enzymes can degrade insect cuticle and gut membranes [11]; our histology confirms extensive midgut damage (Fig. 4).”
- Line 415–420: The sentence about Koch’s postulates is odd: you did isolate S. marcescens from hemolymph, which supports causation. However, Koch’s full postulates require re-inoculation to reproduce disease (which you did, conceptually). It’s fine to mention “consistent with Koch’s postulates”. Perhaps say “We confirmed S. marcescens in the hemolymph of infected larvae (not found in controls), satisfying a criterion of pathogen causality.”
- Line 430–437 (Hemocytes): The discussion on hemocytes has some speculation. For instance, the statement “excess blood cells could disrupt physiology [32,33]” seems tangential. Instead, highlight that the transient surge in plasmatocytes likely reflects activation of cellular immunity, and that the later granulocyte response suggests phagocytic activity. The idea that high immune cell production is energetically costly is valid but should be linked to your data: e.g. “This surge in plasmatocytes (Fig. 3) indicates an active immune response, but it might also divert resources from growth [32,33].” If citing Strand et al., ensure that reference indeed supports this claim. Also, clarify “SP decreased due to chitinase” hypothesis carefully: perhaps say “Since S. marcescens produces chitinase, it could impair spherulocyte function (they are involved in cuticle formation) [34], possibly explaining the drop in SP at 24h.” Without direct evidence, frame as speculation.
- Line 442–450 (Hemocytes cont’d): The text implies granulocyte apoptosis (line 449–450) without evidence. Unless you saw apoptotic bodies, avoid this. Instead: “Granulocytes (GR) are the main phagocytic cells [35], and their increased numbers at 48h suggest recruitment to the infection site. The earlier fluctuation in GR could reflect initial consumption of pathogens or modulation by bacterial factors.”
- Line 454–463 (Midgut damage): The description here largely restates results. Ensure it flows logically: “The midgut peritrophic membrane (PM) usually protects gut cells. In treated FAW, the PM was disrupted by 24–48h, and the epithelial layer became highly disorganized (Fig. 4). This likely allowed S. marcescens to breach the gut barrier. We hypothesize that S. marcescens chitinases contributed to PM degradation, consistent with the observed collapse of midgut structure.” The sentence about reduced gut size improving metabolic efficiency [36] seems out of place: in fact, a smaller gut likely impairs digestion. If [36] is not directly relevant, it could be omitted or rephrased (perhaps “the altered gut morphology would reduce feeding efficiency” if you want to imply slower growth). The discussion of PM proteins [37] is okay but should segue: e.g. “Since the PM isolates food from gut cells [38], its loss would expose epithelium to further damage and infection.” The authors’ note that “reduced secretory capacity of PM” leads to susceptibility is reasonable but ensure it’s tied to a reference.
- Line 471–477 (Vacuolization and regeneration): The citation [40] regarding vacuolization is good. The description of “bulb cytoplasm” and regeneration is hard to parse. Perhaps: “By 72h, we observed numerous cells with large vacuoles, indicative of stress or degeneration. Adjacent tissue showed many new small cells, suggesting active regeneration. These morphological changes imply significant energy investment in midgut repair (consistent with [41]).” If [41] is about energy cost, cite it specifically.
- Line 480–488 (Gut microbiota): The text correctly notes S. marcescens can be both environmental and opportunistic (line 480–483). When saying “Enterococci had the highest contribution in controls” (line 485), cite reference [53] or explain if that came from LEfSe. The sentence “the introduction of ZJ9, certain bacterial groups increased, indicating potential symbiosis or advantage” (line 486–491) is speculative. Perhaps simplify: “After infection, Serratia itself dominated the gut and likely outcompeted or disrupted other symbionts. For example, Enterococcus, common in healthy FAW guts [53], was displaced. Some minor taxa (‘Others’ in Fig. 7) expanded, possibly representing environmental bacteria that proliferate under dysbiosis.” This avoids unsubstantiated “symbiotic relationship” with Serratia.
- Line 499–507 (Microbiome balance): The discussion on gut microbial imbalance is fine, but the line about microbial “reconstruction” after Beauveria [49] and linking to host viability is interesting. Make sure [49] is an FAW-Beauveria reference. The final sentence on “midgut system loses regulatory function” (line 503–505) is speculative; better say “These results suggest that S. marcescens infection disrupts the normal control of the gut microbiome, possibly allowing opportunistic bacteria to flourish.”
- Line 506–515 (Microbiota-host interplay): The points about microbiota as “secondary immune system” [50] and examples with Bt ([51,52]) are relevant. It might fit better earlier when discussing immunity. Here, emphasize how changes in gut flora could worsen infection: e.g. “A disrupted microbiota may weaken gut immunity and nutrient absorption [50]. In some insects, gut symbionts can protect against pathogens, whereas in others they can enhance susceptibility [51,52]. In our case, the bloom of Serratia and loss of Enterococcus may have undermined FAW’s gut defenses.”
- Line 531–537 (Metabolic gene changes): This part is convoluted. Reframe: “Transcriptome data indicate that infection triggered major metabolic shifts. Both treated and control groups at 24 and 48h show higher expression of lipid metabolism genes, but the treatment group in particular showed marked changes relative to control. Concurrently, genes for cell division, synthesis, and tissue repair were upregulated (Table 1 & Fig. 15). This suggests FAW larvae mobilize energy reserves and activate repair processes in response to gut damage, at substantial energetic cost [56].” Ensure [56] (if that’s a reference to energy trade-off) is correct.
- Line 543–552 (Transcriptome and immunity): Good to note that many genes changed (lines 543–546). When you say “immune genes like those related to bacterial infection” were affected, specify which (if any AMPs or PRRs were up/down). The phrase “immune evasion” by Serratia (line 566–569) is interesting but you should base it on observation: did you see downregulation of AMP genes? If yes, say “several antimicrobial peptide genes were unexpectedly downregulated at 48h, which could indicate immune suppression by S. marcescens [62]”. The mention of bees [61,62] is a nice parallel but somewhat tangential; keep it concise.
- Line 575–578 (Serotonin and repair): The finding of serotonin synthesis upregulation is novel. If true, highlight it: “Notably, genes in the serotonin biosynthesis pathway were upregulated, which is intriguing because serotonin can be involved in tissue repair and stress responses [64]. This could relate to the observed histological repair attempts (Fig. 4, 72h).” Just ensure [64] is an appropriate reference (the line implies it is).
- Line 581–584 (Conclusions and Implications): The conclusion rephrases the abstract. You should note limitations: e.g. “These findings are based on laboratory assays; field efficacy and safety of using S. marcescens as a biocontrol agent remain to be tested.” Possibly add one sentence on future work (testing formulations, non-target effects, or combining with other agents).
Author Response

(The authors gave the same response as above.)

Reviewer 3 Report
Comments and Suggestions for Authors
Review
Changes in the gut microbiota, cellular structure, and transcription levels of Spodoptera frugiperda infected by Serratia marcescens
First of all, I consider this paper valuable, however I have some questions about why, and how this paper adding new insight after the paper mentioned (reference 22)? I understand that the effect on midgut tissue is important, but further explanations will be necessary to make a clear distinctions between the two papers.
Please add the sample size at each experiment. How many larvae were used and make is clear the control.
Also make it clear the sample size used for transcriptomic sequencing.
Please provide a separate Data analyses section and explain the data analyses in details.
At results
Why you talking about egg mortality correlations? Why these were not directly assessed. At figure however it looks as you used a positively distributed values for bar charts. Why?
Please explain if standard errors or St. Deviations are presented on bar charts.
Please add more explanations to the figure 3, they must be self-explanatory.
Figures 5, 6, 7, 8 and especially the 9 must have a much more detailed explanations and titles.
Please verify, after figure 14 you have figure 16.
Please increase figures labels especially at 10-16
Figure 17 I completely inexplicable and cannot be read.
Again, for some reason, the reference 22 is not mentioned in Discussion part, however if I understand well, this was the trigger for this research. Please explain.
Comments on the Quality of English LanguageReview
Changes in the gut microbiota, cellular structure, and transcription levels of Spodoptera frugiperda infected by Serratia marcescens
First of all, I consider this paper valuable, however I have some questions about why, and how this paper adding new insight after the paper mentioned (reference 22)? I understand that the effect on midgut tissue is important, but further explanations will be necessary to make a clear distinctions between the two papers.
Please add the sample size at each experiment. How many larvae were used and make is clear the control.
Also make it clear the sample size used for transcriptomic sequencing.
Please provide a separate Data analyses section and explain the data analyses in details.
At results
Why you talking about egg mortality correlations? Why these were not directly assessed. At figure however it looks as you used a positively distributed values for bar charts. Why?
Please explain if standard errors or St. Deviations are presented on bar charts.
Please add more explanations to the figure 3, they must be self-explanatory.
Figures 5, 6, 7, 8 and especially the 9 must have a much more detailed explanations and titles.
Please verify, after figure 14 you have figure 16.
Please increase figures labels especially at 10-16
Figure 17 I completely inexplicable and cannot be read.
Again, for some reason, the reference 22 is not mentioned in Discussion part, however if I understand well, this was the trigger for this research. Please explain.
Author Response

(The authors gave the same response as above.)

Round 2
Reviewer 2 Report
Comments and Suggestions for Authors
No further comments
Reviewer 3 Report
Comments and Suggestions for Authors
Thank you for your corrections and answers. I agree with all your corrections, and thank you for accepting my observations.